# Quadratic Coreset Selection: Certifying and Reconciling Sequence and Token Mining for Efficient Instruction Tuning

**Ziliang Chen**[1], **Yongsen Zheng**[2], **Zhao-rong Lai**[3], **Zhanfu Yang**[4], **Cuixi Li**[5], **Yang Liu**[6], **Liang Lin**[1,6*]

[1]Peng Cheng Laboratory, [2]Nanyang Technological University, [3]Jinan University,
[4]Rutgers University, [5]UNSW, [6]Sun Yat-sen University

## Abstract

Instruction-Tuning (IT) was recently found the impressive data efficiency in post-training large language models (LLMs). While the pursuit of efficiency predominantly focuses on sequence-level curation, often overlooking the nuanced impact of critical tokens and the inherent risks of token noise and biases. Drawing inspiration from bi-level coreset selection, our work provides the principled view of the motivation behind selecting instructions' responses. It leads to our approach Quadratic Coreset Selection (QCS) that reconciles sequence-level and token-level influence contributions, deriving more expressive LLMs with established theoretical result. Despite the original QCS framework challenged by prohibitive computation from inverted LLM-scale Hessian matrices, we overcome this barrier by proposing a novel QCS probabilistic variant, which relaxes the original formulation through re-parameterized densities. This innovative solver is efficiently learned using hierarchical policy gradients without requiring back-propagation, achieving provable convergence and certified asymptotic equivalence to the original objective. Our experiments demonstrate QCS's superior sequence-level data efficiency and reveal how strategically leveraging token-level influence elevates the performance ceiling of data-efficient IT. Furthermore, QCS's adaptability is showcased through its successes in regular IT and challenging targeted IT scenarios, particularly in the cases of free-form complex instruction-following and CoT reasoning. They underscore QCS's potential for a wide array of versatile post-training applications.

## 1 Introduction

Scaling up the sizes of architecture and datasets behind large language models (LLMs) consistently yields the improvement of next-token prediction, elevating the achievement of compressing the broad world knowledge into their parameters [33, 8, 3]. After unsupervised pre-training, task-instruction tuning (IT) [57], or preference modeling approaches such as reinforcement learning from human feedback (RLHF) [4] and direct preference optimization (DPO) [47], can follow to capture the human preference with supervision. Nevertheless, distinct from RLHF and DPO that demand considerable human-preference data to approximate a trustful Bradley-Terry model [7], IT shows otherwise in existing research [68, 18], where curated instruction-response dyads extracted from instruction tuning datasets facilitate comparable instruction following capabilities in LLMs. Such trained model even capably surpasses the performance of LLMs fine-tuned on the complete set of instances. With this regards, efficiently curating diverse and high-quality instruction-response data for post-training LLMs gradually becomes a trend in the community.

---

*Corresponding author.

39th Conference on Neural Information Processing Systems (NeurIPS 2025).

Leading approaches to curate high-quality data mostly select the influential instance as a sequence composed of instruction and response. The conventional next-token prediction loss fairly treats each token in the response, overshadowing the potential profits of critical tokens [32] in the sequence, and the risks in token noises [36, 53] and biases [31] in Zipf's law. In contrast, token-level information recently grasps more attention to enhance the LLM-based fine-tuning performances [65, 66, 67]. Despite so, the existing algorithms are rarely investigated to curate tokens along with sequence subset selection due to the prohibitive computational burden.

To rescue IT from the data-efficient dilemma, we dived into the principled analysis between sequence and token selection derived from the spirit of bi-level coreset selection [6]. The methodology inspired from bi-level optimization, yields the objective with small subset to train a model consistent with that one trained with the entire set. It is observed that the common goals of most existing sequence selection and token selection methods can be formally interpreted by the bi-level coreset selection problem with sequences or tokens as their selected instances. With such regards, we propose *quadratic coreset selection* (QCS) to optimize the subset weights jointly determined by the sequence-level and the token-level influence contributions. It can be proved that the solution set of QCS incorporates all LLMs elicited by the sequence-level and token-level selection, verifying the technical benefit to reconcile sequence and token mining for IT.

Despite achieving the best of both worlds, QCS is hampered by the costly token-level computation from the traditional solver using the inverses of Hessian matrix with the dimensions in the parameter size of LLMs. To this end, our work circumvents the unaffordable computation by a newly proposed QCS probabilistic solver, relaxing the original QCS formulation through a re-parameterized density to assign the quadratic coreset weights across tokens and sequences with the sampling policies on their own. We certify the asymptotic equivalence between the next-token prediction empirical risk trained with the subsets drawn from QCS and its probabilistic parameterized variant and more importantly, the solver can be learned by hierarchical policy gradients without back-propagation, served by the provable convergence under mild conditions.

Our experimental results demonstrate the efficacy of our QCS algorithm from two perspectives: (1). we evaluate its instruction-following ability with the sequence-level data efficiency, showcasing the generalization with limited data; (2) we analyze the token-level benefits and inner-loop adjustments within QCS, revealing the cause of its success. QCS is further evidenced through its reconfiguration for targeted IT ([60]), a challenging transfer learning setup for data-efficient training for LLMs; and its replay-based variant to overcome the catastrophic-forgetting challenges in continual IT (Appendix.D). They jointly justify its potential to suit more versatile scenarios of post-training in practice.

## 2 Related Work

**Data efficiency in instruction tuning (IT).** Using high-calibrated IT data has been demonstrated to significantly enhance the performance of LLM. IT datasets can be mostly categorized into two threads, *i.e.*, task-oriented datasets derived from established NLP tasks [56, 48, 58] and open-ended instruction following datasets that encompass a broad spectrum of subject matter [50, 16, 35, 62, 43, 17]. Among recent research, the priority of data quality and diversity instead of quantity, enables more efficacious cultivation of instruction-following ability [13, 12, 9, 19, 38, 10, 11, 30, 37, 69]. More recently, [60, 20] proposed using efficient estimation [44, 25] of influence functions [46] to select the instruction sequences with key effects in IT. They are technically connected with our methodology.

**Curating tokens to learn LLMs.** Diverse trends of LLM-based training involve exploration in token-level knowledge, including pre-training [29, 39] and fine-tuning [65, 66, 67]. Different types of tokens, as investigated in previous studies [39, 24, 36, 53, 32, 34, 1, 2], may hold diverse influences to LLMs. Their evidences imply the potential improvement of IT if sequence-level IT data selection can be merged with token-level knowledge.

**Coreset-based data subset selection.** The process of curating high-quality data can be reframed as a coreset selection problem [51, 26, 61]. The methodology endeavors to identify a weighted subset of instances whose objective function, when utilized, yields the consistent performance to that trained through the usage of the entire datasets. Early efforts investigate the solver of specific learning algorithms, *e.g.*, $k$-means [22] and GMMs [42], *etc*, hardly combined with arbitrary models. Recently, [6] unifies black-box parameter learning with coreset selection into a discrete bi-level optimization

paradigm:

$$\min_{\boldsymbol{v}\in\mathcal{C}} \frac{1}{n}\sum_{i=1}^{n}\mathcal{L}_i(\theta^*(\boldsymbol{v})), \text{s.t. } \theta^*(\boldsymbol{v})\in\arg\min_{\theta\in\Theta}\frac{1}{K}\sum_{i=1}^{n}v_i\mathcal{L}_i(\theta(\boldsymbol{v})) \tag{1}$$

where $\mathcal{L}_i$ denotes a training error of the $i$-th sample, $\boldsymbol{v}\in\mathcal{C}$ indicates the binary weights of training set valued in a feasible region $\mathcal{C}=\{\boldsymbol{v}:v_i=0/1, ||\boldsymbol{v}||_0\leq K\}$, with a integer $K>0$ to control the coreset size. The coreset $\boldsymbol{v}$ searched by (1) enables the integration with many applications. However, its greedy algorithm based on cone constrained optimization [40] needs to compute a Hessian inverse with the parameter dimension, poorly scaling to big models. Two threads of alternatives refer to matrix approximation [45] or probabilistic derivation [70], respectively. The first is closely relevant with influential functions [46] employed by LESS and the second resorts to a Bernoulli density to re-parameterize (1), laying the background of our algorithm.

## 3  Sequence or Token Mining for Instruction Tuning: A Coreset Viewpoint

Let first review the IT procedure as the next-token prediction with cross-entropy losses as follows:

$$\mathcal{L}_{\mathsf{IT}}(\theta) := -\frac{1}{M}\sum_{i=1}^{M}\Big(\frac{1}{N_i}\sum_{j=1}^{N_i}\log P(x_{i,j}|\boldsymbol{x}_{i,<j},\boldsymbol{z}_i;\theta)\Big) = \frac{1}{M}\sum_{i=1}^{M}\Big(\frac{1}{N_i}\sum_{i=1}^{N_i}\mathcal{L}_{i,j}(\theta)\Big) \tag{2}$$

where we represent $\mathcal{L}_{i,j} = -\log P(x_{i,j}|\boldsymbol{x}_{i,<j},\boldsymbol{z}_i;\theta)$, then (2) holds the equivalent form $\mathcal{L}_{\mathsf{IT}}(\theta)$ $= \frac{1}{M}\sum_{i=1}^{M}\Big(\frac{1}{N_i}\mathcal{L}_{i,j}(\theta)\Big)$; $M$ indicates the number of instruction-response pairs, then given the instance $i$, LLMs with the pre-trained parameter $\theta$ receives the instruction $\boldsymbol{z}_i$ along with the observed response content $\boldsymbol{x}_{i,<j}$, to predict the $j$-th token $x_{i,j}$. (2) is a simple yet powerful paradigm for the first step of most existing post-training, while more evidences in leading research demonstrated that we (2) does not require the whole dataset: post-training with high-quality instruction responses or the emphasis of critical tokens might elicit the consistent, even more powerful models than the original.

As discussed previously, suppose that some data-efficient IT algorithm aims to select $K$ instruction-response pairs ($K<M$) to chase for the optima with regards to (2). Hence given the selected instances denoted by the indicator set $\boldsymbol{v}$ lies in $\mathcal{C}_{\mathsf{seq}}=\{\boldsymbol{v}:v_i=0/1, ||\boldsymbol{v}||_0\leq K\}$, the motivation behind refers to the bi-level objective

$$\min_{\boldsymbol{v}\in\mathcal{C}_{\mathsf{seq}}}\frac{1}{M}\sum_{i=1}^{M}\Big(\frac{1}{N_i}\sum_{i=1}^{N_i}\mathcal{L}_{i,j}(\theta^*(\boldsymbol{v}))\Big), \text{ (sequence level coreset)}$$

$$\text{s.t., } \theta^*(\boldsymbol{v})\in\arg\min_{\theta\in\Theta}\frac{1}{K}\sum_{i=1}^{M}v_i\Big(\frac{1}{N_i}\sum_{j=1}^{N_i}\mathcal{L}_{i,j}(\theta(\boldsymbol{v}))\Big), \tag{3}$$

where $\theta^*(\boldsymbol{v})$ represents the LLM trained with instances identified by $\boldsymbol{v}$, seeking to achieve the original goal of $\mathcal{L}_{\mathsf{IT}}(\theta)$. So obviously, (2) exactly refers to the bi-level coreset selection in (1) with regards to instruction-response sequences as selected instances.

Resembling the similar analysis, token-selection algorithms that attempt to maintain the consistent performance with the original IT goal, also elicits the token-level corset selection

$$\min_{\boldsymbol{v}\in\mathcal{C}_{\mathsf{tkn}}}\frac{1}{M}\sum_{i=1}^{M}\Big(\frac{1}{N_i}\sum_{i=1}^{N_i}\mathcal{L}_{i,j}(\theta^*(\boldsymbol{v}))\Big), \text{ (token level coreset)}$$

$$\text{s.t., } \theta^*(\boldsymbol{v})\in\arg\min_{\theta\in\Theta}\frac{1}{M}\sum_{i=1}^{M}\Big(\frac{1}{K_i}\sum_{j=1}^{N_i}v_{i,j}\mathcal{L}_{i,j}(\theta(\boldsymbol{v}))\Big) \tag{4}$$

with $\mathcal{C}_{\mathsf{tkn}}=\cap\mathcal{C}_i$ and $\forall i\in[M], \mathcal{C}_i=\{\boldsymbol{v}_i:v_{i,j}=0/1, ||\boldsymbol{v}_i||_0\leq K_i\}$ ($K_i$ indicates the cap of tokens selected in the sequence $i$).

**Motivation: the dilemma between sequence-level and token-level coreset selection.** The pursuit of data efficiency in instruction tuning through coreset selection, whether at the sequence level ( 3) or token level ( 4), reveals an inherent tension in methodology. Specifically,

- **Sequence-level coreset overlooks token information.** (3) encourages LLMs to follow task instructions with the instruction-response coreset selected in IT datasets. It is consistent with other data-curating methods, treats tokens equally per sequence. Yet it easily suffers from token noises and bias since critical tokens with factuality and others are fair in a sequence.

- **Token-level coreset is computationally expensive and memory-hungry.** The optimal token-level coreset in (4) serves a lower bound of (3) in the same feasible regions. Despite this superiority, (4) is hardly achieved in practice due to its prohibitive computation with the token amount quadratically larger than sequential instances. Moreover, the data efficiency by token selection is a mirage: suppose the $j$-th token $x_{i,j}$ is pruned by (4). The token $x_{i,j}$ remains required as a part of the conditioned response context $\boldsymbol{x}_{i,<j+k}$ in $P(x_{i,j+k}|\boldsymbol{x}_{i,<j+k}, \boldsymbol{z}_i; \theta)$ ($k \in \mathbb{N}_+$). The retention for contexts negates practical memory saving.

# 4 Proposed Approach: Quadratic Coreset Selection

In the previous section, we note that common goals of massive sequence or token selection algorithms typically refer to (3,4) that remain the conflict. While their disadvantages are precisely the advantages of each other, suggesting that the limitations can be effectively mitigated by their synergistic integration. To this end, we unify them to derive the novel *quadratic coreset selection* (QCS) paradigm to obtain the best of both worlds:

$$
\min_{\boldsymbol{w}\in\mathcal{C}_{\mathsf{seq}},\boldsymbol{v}\in\mathcal{C}_{\mathsf{tkn}}} \mathcal{L}(\theta^*(\boldsymbol{w},\boldsymbol{v})) = \frac{1}{M}\sum_{i=1}^{M}\Big(\frac{1}{N_i}\sum_{i=1}^{N_i}\mathcal{L}_{i,j}(\theta^*(\boldsymbol{w},\boldsymbol{v}))\Big),
$$

$$
\text{s.t.}, \theta^*(\boldsymbol{v}) \in \arg\min_{\theta\in\Theta}\hat{\mathcal{L}}(\theta(\boldsymbol{w},\boldsymbol{v})), \ \hat{\mathcal{L}}(\theta(\boldsymbol{w},\boldsymbol{v})) = \frac{1}{K}\sum_{i=1}^{K}w_i\Big(\frac{1}{K_i}\sum_{j=1}^{N_i}v_{i,j}\mathcal{L}_{i,j}(\theta(\boldsymbol{w},\boldsymbol{v}))\Big).
$$

(5)

$\theta^*(\boldsymbol{w},\boldsymbol{v})$ indicates the parameter optimized with regards to $\boldsymbol{w}$ to select the instruction-response sequences and $\boldsymbol{v}$ to select the tokens in each sequence. It leads to the optimization objective with $\boldsymbol{w}$ and $\boldsymbol{v}$ before using them in IT.

**Theorem 1.** *(**Expressiveness of QCS**) Suppose that $\Omega(\theta_{\mathsf{seq}})$, $\Omega(\theta_{\mathsf{tok}})$ denotes the solution sets of $\theta_{\mathsf{seq}}$, $\theta_{\mathsf{tok}}$ in the sequence-level coreset selection (3) and token-level coreset selection (4), respectively; and $\Omega(\theta_{\mathsf{qcs}})$ denotes the solution sets of $\theta_{\mathsf{qcs}}$ in (5). It holds $\Omega(\theta_{\mathsf{qcs}}) \subset \Omega(\theta_{\mathsf{seq}}) \cap \Omega(\theta_{\mathsf{tok}})$.*

The aforementioned theory demonstrates that, fine-tuning LLMs $\theta$ with instructions with either the response sequences selected by (3) or the next-token prediction losses screened by (4), potentially under-perform due to resulting LLMs with the lower expressiveness. Observed that most existing sequence or token selection nails down to the common goal in (3)(4), the theoretical result verifies the technical merit of the marriage between sequence and token mining for IT. In our Appendix.A, using the typical algorithm for original bi-level coreset selection, we further unveils that the next-token prediction losses in QCS are simultaneously associated with *sequence-level empirical influence* and *token-level empirical influence*. It demonstrates the deep connection between QCS and LESS [60].

Despite reflecting the compound influence effect, traditional algorithms for bi-level coreset are hardly transferred into QCS due to their need of the inverse of Hessian matrix $\left(\frac{\partial f(\theta(\boldsymbol{w}^*,\boldsymbol{v}^*))}{\partial\theta^\top\theta}\right)^{-1}$, which in reality is in a colossal size that grow with the number of LLM's parameters. As an inner-loop stage of bi-level optimization objective, computing the inverse is too compromising to suit the regime. Some alternative refers to find its approximation, which have been employed by influential-based approaches, *e.g.*, LESS [60]. In contrast, we propose a new variant objective relaxed by continual value with probability.

# 5 Probabilistic Re-parameterized Solver for Quadratic Coreset

## 5.1 Probabilistic Re-parameterized Solver with Asymptotic Equivalence

More specifically, we re-conceptualize (5) by the probabilistic reparameterization, where the original discrete coreset weights $\boldsymbol{v}$ and $\boldsymbol{w}$ were turned into binary random variables that underlie a Bernoulli distribution family, *i.e.*,

$$
\forall i \in [K], \ w_i \sim \mathsf{Bern}(\mathsf{w}_i); \ \forall j \in [K_i], \ v_{i,j} \sim \mathsf{Bern}(\mathsf{w}_i\mathsf{v}_{i,j}),
$$

(6)

where $\forall i \in [K]$, $\mathsf{w}_i \in [0,1]$ and $\forall j \in [K_i]$, $\mathsf{w}_i \mathsf{v}_{i,j} \in [0,1]$ indicate the hyper-parameters to develop the Bernoulli distribution family. Assume that the sequence variables underlying $\text{Bern}(\mathsf{w}_i)$ ($\forall i \in K$) are independent, then the distribution function of sequence-level coreset $\boldsymbol{w}$ could be achieved by $p(\boldsymbol{w}|\mathsf{w}) = \prod_{i=1}^{K} (w_i)^{\mathsf{w}_i}(1-w_i)^{(1-\mathsf{w}_i)}$. Similarly, suppose that $\forall i \in [K]$, we got their discrete weights $\boldsymbol{w}$ drawn from $p(\boldsymbol{w}|\mathsf{w})$, then the distribution of the token-level coreset $\boldsymbol{v}$, *i.e.*, $p(\boldsymbol{v}|\boldsymbol{w} \odot \mathsf{v})$ can be formulated as

$$p(\boldsymbol{v}|\boldsymbol{w} \odot \mathsf{v}) = \prod_{i=1}^{K} \Big( \prod_{j=1}^{K_i} (v_{i,j})^{w_i \mathsf{v}_{i,j}} (1-v_{i,j})^{(1-w_i \mathsf{v}_{i,j})} \Big). \tag{7}$$

Note that $p(\boldsymbol{v}|\boldsymbol{w} \odot \mathsf{v})$ depends on the coreset $\boldsymbol{w}$ drawn from $p(\boldsymbol{w}|\mathsf{w})$. It is aligned with the fact that the token-level coreset selection with regards to $\boldsymbol{v}$ is only available in the chosen sequences with regards to $\boldsymbol{w}$.

With the coreset densities $p(\boldsymbol{w}|\mathsf{w})$ and $p(\boldsymbol{v}|\boldsymbol{w} \odot \mathsf{v})$, the coreset sizes defined by $\mathcal{C}_{\text{seq}}$ and $\mathcal{C}_{\text{tkn}}$ can be controlled via their probability masses $\mathbf{1}^\top \mathsf{w}$ and $\mathbf{1}_{\mathcal{C}_i}^\top \mathsf{v}$ because of $\mathbb{E}_{\boldsymbol{w} \sim p(\boldsymbol{w}|\mathsf{w})}||\boldsymbol{w}||_0 = \sum_{i=1}^{K} \mathsf{w}_i$ and $\mathbb{E}_{\boldsymbol{v} \sim p(\boldsymbol{v}|\boldsymbol{w} \odot \mathsf{v})} ||\boldsymbol{v}||_0 = \sum_{i=1}^{K} \mathbb{E}_{\boldsymbol{v} \sim p(\boldsymbol{v}|\boldsymbol{w} \odot \mathsf{v})} ||\boldsymbol{v} \odot \mathbf{1}_{\mathcal{C}_i}||_0 = \sum_{i=1}^{K} \sum_{j=1}^{K_i} ||\mathsf{v}_{i,j}||_0$. It turns the feasible regions $\mathcal{C}_{\text{seq}}$, $\mathcal{C}_{\text{tkn}}$ into their continuous counterparts

$$\begin{aligned} \mathcal{C}_{\text{seq}} &= \{\mathsf{w} : \mathbf{0} \preceq \mathsf{w} \preceq \mathbf{1}, ||\mathsf{w}||_1 \leq K\}; \mathcal{C}_{\text{tkn}} = \cap_{k \in [K]} \mathcal{C}_k, \forall i \in [K], \\ \mathcal{C}_i &= \{\mathsf{v} : \mathbf{0} \preceq \mathsf{v} \odot \mathbf{1}_{\mathcal{C}_i} \preceq \mathbf{1}, \mathbf{1}_{\mathcal{C}_i}^\top \mathsf{v} \leq K_i\}, \end{aligned} \tag{8}$$

Hence the original QCS objective (5) can be eventually transformed into

$$\min_{\mathsf{w} \in \mathcal{C}_{\text{seq}}, \mathsf{v} \in \mathcal{C}_{\text{tkn}}} \Phi(\mathsf{w}, \mathsf{v}) = \mathbb{E}_{p(\boldsymbol{w}|\mathsf{w})p(\boldsymbol{v}|\boldsymbol{w} \odot \mathsf{v})} \mathcal{L}(\theta^*(\boldsymbol{w}, \boldsymbol{v})), \ \text{s.t.} \ \theta^*(\boldsymbol{v}) \in \arg\min_{\theta \in \Theta} \hat{\mathcal{L}}(\theta(\boldsymbol{w}, \boldsymbol{v})) \tag{9}$$

The probabilistic reparameterization for QCS offers many technical benefits: first, (9) encourages the coreset optimized based on the learnable parameters of the Bernoulli distribution family efficiently by gradients (as demonstrated in the next section); second, the stochastic sampling provides the lower bound of the original, leading to a better optimization objective; third, the sparsity constraints on $\mathsf{w}$ and $\mathsf{v}$ make their optimal value in either $0$ or $1$, *i.e.*, approach the solution of (5). Our following theorem essentially supports the feasibility to replace (5) by (9) given training data sufficiently large.

**Theorem 2.** *(Asymptotic equivalence between QCS and its probabilistic re-parameterized solver) Under the assumptions in Appendix.A, as $n \to \infty$ the difference between the deterministic objective and the probabilistic (expected) objective converges at the standard Monte Carlo rate. More precisely,*

$$\sqrt{n} \left( \mathcal{L}(\theta^*(\boldsymbol{w}, \boldsymbol{v})) - \Phi(\mathsf{w}, \mathsf{v}) \right) \xrightarrow{d} \mathcal{N}(0, \sigma^2), \tag{10}$$

*for some $\sigma^2 > 0$. Equivalently,*

$$|\mathcal{L}(\theta^*(\boldsymbol{w}, \boldsymbol{v})) - \Phi(\mathsf{w}, \mathsf{v})| = O_p \left( \frac{1}{\sqrt{n}} \right),$$

*where $O_p(\cdot)$ denotes convergence in probability.*

### 5.2 Solver Learning by Hierarchical Policy Gradients

Distinct from (5) directly chasing for the optimal coresets $\boldsymbol{w}^*$ and $\boldsymbol{v}^*$, the probabilistic QCS solver achieves IT with the coresets obtained by drawing the sequences $\boldsymbol{w}$ and tokens $\boldsymbol{v}$ from the Bernoulli distribution family (6) with the optimal hyper-parameter $\mathsf{w}^*$ and $\mathsf{v}^*$. The deterministic coreset optimization with $\boldsymbol{w}$ and $\boldsymbol{v}$ is non-trivial because their gradient estimators contain their implicit differentiation. Instead, its probabilistic solver transfer the problems into learning the hyper-parameters $\mathsf{w}$ and $\mathsf{v}$. If we consider the samplers $p(\boldsymbol{w}|\mathsf{w})$ and $p(\boldsymbol{v}|\boldsymbol{w} \odot \mathsf{v})$ through a lens of sampling policy, their optimization can be achieved by our hierarchical policy gradient estimators (HPGEs), *i.e.*,

$$\begin{aligned} \nabla_{\mathsf{w}} \Phi(\mathsf{w}, \mathsf{v}) &\approx \nabla_{\mathsf{w}} \Big( \int \mathcal{L}(\theta^*(\boldsymbol{w}, \boldsymbol{v})) p(\boldsymbol{w}|\mathsf{w}) p(\boldsymbol{v}|\boldsymbol{w} \odot \mathsf{v}) d\boldsymbol{w} d\boldsymbol{v} \Big) \\ &= \int \mathcal{L}(\theta^*(\boldsymbol{w}, \boldsymbol{v})) \frac{\nabla_{\mathsf{w}} p(\boldsymbol{w}|\mathsf{w})}{p(\boldsymbol{w}|\mathsf{w})} p(\boldsymbol{w}|\mathsf{w}) p(\boldsymbol{v}|\boldsymbol{w} \odot \mathsf{v}) d\boldsymbol{w} d\boldsymbol{v} \\ &= \mathbb{E}_{p(\boldsymbol{w}|\mathsf{w}) p(\boldsymbol{v}|\boldsymbol{w} \odot \mathsf{v})} \big[ \mathcal{L}(\theta^*(\boldsymbol{w}, \boldsymbol{v})) \nabla_{\mathsf{w}} \ln p(\boldsymbol{w}|\mathsf{w}) \big], \end{aligned} \tag{11}$$

and

$$\nabla_{\mathbf{v}}\Phi(\mathbf{w}, \mathbf{v}) \approx \nabla_{\mathbf{v}}\Big( \int \mathcal{L}(\theta^*(\boldsymbol{w}, \boldsymbol{v}))p(\boldsymbol{w}|\mathbf{w})p(\boldsymbol{v}|\boldsymbol{w} \odot \mathbf{v})d\boldsymbol{w}d\boldsymbol{v}\Big)$$

$$= \int \mathcal{L}(\theta^*(\boldsymbol{w}, \boldsymbol{v}))p(\boldsymbol{w}|\mathbf{w})\frac{\nabla_{\mathbf{v}}p(\boldsymbol{v}|\boldsymbol{w} \odot \mathbf{v})}{p(\boldsymbol{v}|\boldsymbol{w} \odot \mathbf{v})}p(\boldsymbol{v}|\boldsymbol{w} \odot \mathbf{v})d\boldsymbol{w}d\boldsymbol{v} \qquad (12)$$

$$= \mathbb{E}_{p(\boldsymbol{w}|\mathbf{w})p(\boldsymbol{v}|\boldsymbol{w}\odot\mathbf{v})}\big[\mathcal{L}(\theta^*(\boldsymbol{w}, \boldsymbol{v}))\nabla_{\mathbf{v}}\ln p(\boldsymbol{v}|\boldsymbol{w} \odot \mathbf{v})\big].$$

Note that $\mathcal{L}(\theta^*(\boldsymbol{w}, \boldsymbol{v}))\nabla_{\mathbf{w}}\ln p(\boldsymbol{w}|\mathbf{w})$ and $\mathcal{L}(\theta^*(\boldsymbol{w}, \boldsymbol{v}))\nabla_{\mathbf{v}}\ln p(\boldsymbol{v}|\boldsymbol{w} \odot \mathbf{v})$ are unbiased stochastic estimators of $\nabla_{\mathbf{w}}\Phi(\mathbf{w}, \mathbf{v})$ and $\nabla_{\mathbf{v}}\Phi(\mathbf{w}, \mathbf{v})$. It encourages the forward gradient calculation to estimate $\nabla_{\mathbf{w}}\Phi(\mathbf{w}, \mathbf{v})$, $\nabla_{\mathbf{v}}\Phi(\mathbf{w}, \mathbf{v})$ without the extra backward operation. Moreover, $\nabla_{\mathbf{w}}\Phi(\mathbf{w}, \mathbf{v})$ and $\nabla_{\mathbf{v}}\Phi(\mathbf{w}, \mathbf{v})$ can be independently estimated despite the dependency from sequence-level policy to token-level policy. They sufficiently ease the calculation of the coreset selection process.

**Projected hierarchical policy gradients.** In terms of the constraints refined to QCS probabilistic solver in (8), we may simply apply them in HPGE by projected stochastic gradient decent, in order to constrain the hyper-parameters optimized in the proper range:

$$\mathbf{w} \leftarrow \mathcal{P}_{\mathcal{C}_{\text{seq}}}\big(\mathbf{w} - \eta_1\mathcal{L}(\theta^*(\boldsymbol{w}, \boldsymbol{v}))\big)\nabla_{\mathbf{w}}\ln p(\boldsymbol{w}|\mathbf{w})$$
$$\mathbf{v} \leftarrow \mathcal{P}_{\mathcal{C}_{\text{tkn}}}\big(\mathbf{v} - \eta_2\mathcal{L}(\theta^*(\boldsymbol{w}, \boldsymbol{v}))\big)\nabla_{\mathbf{v}}\ln p(\boldsymbol{v}|\boldsymbol{w} \odot \mathbf{v}). \qquad (13)$$

With this observation, we can directly obtain the close-form solution to the projection, which yields a very efficient process to update $\mathbf{w}$ and $\mathbf{v}$ for our coreset sampler (see our Appendix.A). More importantly, we further prove that if the inner-loop optimization $\hat{\mathcal{L}}(\theta(\boldsymbol{w}, \boldsymbol{v}))$ with the sampled coresets $\boldsymbol{w},\boldsymbol{v}$ converges, the outer-loop algorithm for our hierarchical coreset selection policy also converge as the projected stochastic gradient decent [23]:

**Proposition 3.** *[Convergence of hierarchical-policy coreset sampler (Informal)] Under the mild conditions of $\Phi(\mathbf{w}, \mathbf{v})$ and the step sizes $\eta_1$, $\eta_2$, the average of the expectation of the norms for sequence-level and token level projected gradients,* i.e.

$$\Big|\Big|\frac{1}{\eta_1}\Big(\mathbf{w}_t - \mathcal{P}_{\mathcal{C}_{\text{seq}}}\big(\mathbf{w}_t - \eta_1\nabla_{\mathbf{w}}\Phi(\mathbf{w}_t, \mathbf{v}_t)\big)\Big)\Big|\Big|_2; \quad \Big|\Big|\frac{1}{\eta_2}\Big(\mathbf{v}_t - \mathcal{P}_{\mathcal{C}_{\text{seq}}}\big(\mathbf{v}_t - \eta_2\nabla_{\mathbf{v}}\Phi(\mathbf{w}_t, \mathbf{v}_t)\big)\Big)\Big|\Big|_2, \quad (14)$$

*both converge to 0 as $T \to \infty$.*

## 6    Algorithms

### 6.1    Practical Implementation of Inner-Loop Update

Whatever variants of QCS require the optimization process in the inner-outer loop alternation, where the inner loop achieves IT with coreset samples. In contrast with the efficiency in the outer loop, learning $\theta$ in the inner-loop conveys the heavy computation because $\theta$ denotes the parameter of LLMs. To this end, we further discuss the possible improvement to the inner loop.

**Elastic token coreset.** In the QCS inner loop, $\hat{\mathcal{L}}(\theta(\boldsymbol{w}, \boldsymbol{v}))$ is optimized via the sampled binary weights of $\boldsymbol{w},\boldsymbol{v}$. Distinct from sequences, the token selection does not earn any memory efficiency as we previously discussed, on the contrary, the sparsity of $\boldsymbol{v}$ may cause unpredictable variance to minimize the token losses. In this regards, after the optimized hyper-parameters select the coresets $\boldsymbol{w},\boldsymbol{v}$, we re-weight each value of token-level coreset $\boldsymbol{v}$ from $\{0, 1\}$ into $\{\epsilon, 1 - \epsilon\}$, where $\epsilon \in (0, 1)$ denotes the elastic parameter close to 0.

**Parameter-efficient fine-tuning (PEFT).** During the alternative optimization, the IT process in the inner loop actually serves for learning the solver parameters $\mathbf{v}$, $\mathbf{w}$. With this finding, it is unnecessary to fine-tune the full parameters in the LLM until $\mathbf{v}$, $\mathbf{w}$ converge. In lieu of full-parameter IT, we employ low-rank adaptation (LoRA) [28] to facilitate the IT process in the inner loop. The LoRA adapter is trained with the full set of IT data. The process is executed along with the update of $\mathbf{v}$, $\mathbf{w}$.

### 6.2    Transfer instruction-tuning with quadratic coresets

QCS implemented by the gradient update just provide the pre-selection probability of $\mathcal{D}_{\text{train}}$, while does not incorporate any information from $\mathcal{D}_{\text{val}}$. To enable the task transfer of training sequence-token selection from $\mathcal{D}_{\text{train}}$ to $\mathcal{D}_{\text{val}}$, we encourage the instance-level Bernoulli hyper-parameters $\text{w}_i$ in $\mathbf{w}$

and $\mathsf{v}_{i,j}$ in **v** generated by language model itself. Specifically, we augment the pre-trained LLM with frozen parameter $\overline{\theta}$ with different LoRA modules with regards to **w** and **v**, then produce binary classifier heads to generate $\mathsf{w}_i$ and $\mathsf{v}_{i,j}$, respectively, as follows:

$$\mathsf{w}_i = \mathsf{LLM}_{\overline{\theta},\phi}(\boldsymbol{x}_i, \boldsymbol{z}_i); \mathsf{v}_{i,j} = \mathsf{LLM}_{\overline{\theta},\psi}(\boldsymbol{x}_{i,<j}, \boldsymbol{z}_i), \text{ s.t.} \forall (\boldsymbol{x}_i, \boldsymbol{z}_i) \in \mathcal{D}_{\mathsf{train}}. \tag{15}$$

These uncoupled sub-nets $\phi$ and $\psi$ are trained with $\mathcal{D}_{\mathsf{train}}$ using the selection labels indicated by the optimal $\mathsf{w}_i^*$ and $\mathsf{v}_{i,j}^*$ obtained by the gradients (11, 12), then simple entropy-based test-time adaptation [54] would be employed to fast update their selection probability in $\mathcal{D}_{\mathsf{val}}$.

The vanilla pipeline of QCS and its transfer learning variant have been shown in Appendix.A.

## 7 Experiments

In this section, we evaluate QCS based on targeted IT and regular IT. Appendix.D also illustrate the empirical study on continual IT. They verify the versatility of QCS.

### 7.1 Targeted Instruction Following

#### 7.1.1 Experimental Setup

Targeted IT denotes the problem that based on a handful samples $\mathcal{D}_t$ for each target IT task $t$, we accordingly select a small proportion of source IT data to adapt the task. The experimental setup is coordinated with the setup of **LESS**, where 5% training sequences are selected to update the LLMs for each task. **QCS** follows its transfer learning setup. More details found in Appendix.C.

**Training Datasets.** Our fine-tuning resembles the setup in LESS [60]. Specifically, we adopt the IT datasets outlined by [55] that includes two branches. (1) the datasets derived from existing collections, such as **FLAN V2** [41] and **COT** [59], and (2) open-ended generation datasets featuring human-authored responses, including **DOLLY** [16] and **OPEN ASSISTANT** [35]. The diverse corpus includes 270K sequences in a wide range of formats and reasoning tasks.

**Evaluation Datasets.** Our evaluation leverages four comprehensive benchmarks: **MMLU**, **TYDIQA**, **BBH** [49], and **GSM8K** [15]. Each evaluation set has multiple tasks while samples in each task are split into $\mathcal{D}_{\mathsf{val}}$ and $\mathcal{D}_{\mathsf{test}}$, where $\mathcal{D}_{\mathsf{val}}$ only contains few-shot demonstration sequences while we need to use $\mathcal{D}_{\mathsf{val}}$ to select the influential samples in the training datasets, specifically available to this task, then use them to training LLM in order to well perform in $\mathcal{D}_{\mathsf{test}}$.

**Base LLMs for Data Selection and Training.** Before introducing baselines, we specify the LLM used to evaluate the baselines for a comparison. Concretely, we employ the popular **LLAMA-2** series [52] to evaluate the main results across evaluated baselines for MMLU, TYDIQA, BBH; yet select **LLAMA-2-7B** and **LLAMA-3-8B** as the base LLMs for GSM8K. Finally, we employ **PYTHIA** [5], a small LLM series to evaluate the adaptability of algorithm under scaling law.

**Baselines.** The motivation of QCS rises from the data efficiency of IT. To this, beyond the original IT baseline **FULL** using full IT data, we also employ another two data selection approaches invented to choose key sequences for IT. Specifically, we adopt **RAND** as the simplest baseline to randomly choose some proportion of sequences for IT, and **LESS** [60], to the best of our knowledge, the most competitive baseline in targeted IT problem.

#### 7.1.2 Main Results.

We present main results of different baselines via diverse LLAMA's family models in Figure.1; and a thorough evaluation between LESS and QCS across Pythia series in Table.1,2.

**QCS *v.s.* LESS: the response size matters.** The results of four baselines across four benchmarks using different foundation models are observed in Fig.1. Though targeted data selection only exceeds FULL in 1 of 4 benchmarks, they significantly outperform Rand in all cases. In MMLU and TYDIQA, QCS is very competitive yet remains slightly inferior to LESS. It is possibly due to the short response size of evaluation set: LLM only need to select the options in the context. In BBH and GSM-8K, with their answers in a free-form long responses, QCS outperform LESS significantly. In the modern LLAMA-3-8B, it even outperforms the LESS baseline by over 12 points (56.7) , proving its sophisticated ability to prioritize the most impactful training instances.

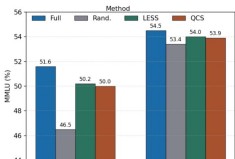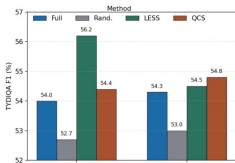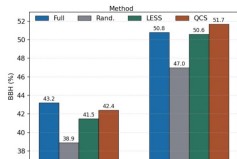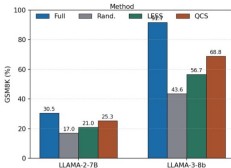

Figure 1: Results of LESS (selected with target model's gradient store) and QCS (pre-trained to obtain , then fined-tuned with ) on LLAMA-2-7B and LLAMA-2-13B. Full denote full dataset, and otherwise we select 5% of the data with random selection (Rand), LESS and QCS. Bold numbers denotes the best performing selected subset. Underlined numbers denote that the selected subset outperforms the full dataset. Numbers in the parentheses are standard deviations.

Table 1: Performance comparison of using different variants of Pythia models for data-efficient targeted IT.

|  | Baselines | Warmup training | Feature gradient/ Adaptive policy | Data selection |
|---|---|---|---|---|
| Time cost | LESS | ~6 | ~48 | <1 min |
|  | QCS | ~30 | <1 | <1 min |
| Memory cost | LESS | 0 | 17.7G | 0 |
|  | QCS | 176.9M | 0 | 176.9M |

Table 2: Performance comparison of using different variants of Pythia models for data-efficient targeted IT.

|  | w/o FT | RAND | LESS Sequence level | QCS | QCS Token level | FULL |
|---|---|---|---|---|---|---|
| FT data | 0% | 5% | 5% | 5% | 5% | 100% |
| PYTHIA-14m | 0.0 | 0.0 | 0.0 | 0.0 | **0.1** | 0.0 |
| PYTHIA-410m | 3.8 | 7.8 | **9.4** | 8.0 | **9.4** | 8.6 |
| PYTHIA-1b | 9.2 | 12.3 | **17.2** | 14.6 | **17.4** | 13.7 |
| PYTHIA-6.9b | 17.1 | 27.2 | 30.9* | 27.2 | 31.6 | **35.5** |
| PYTHIA-12b | 17.4 | 14.9 | 32.6* | 29.8 | 32.3 | **33.2** |
| Average | 9.5 | 12.4 | 18.0 | 15.6 | **18.2** | 18.2 |
| LLAMA-2-7B | 46.4 | 52.7 | **56.2** | 54.4 | **56.6** | 54.0 |

**Rivaling LESS with less computation.** The empirical results presented in Table 1 demonstrate the computation-economical superiority as a sequence subset selection methodology compared with LESS. As we know, instead of **LESS** using feature gradient to select sequences, **QCS** only uses Bernoulli distribution to sample the coreset weights. It saves the computation and memory cost to achieve sequence-token joint selection, while reserve the more impressive performance than LESS.

**Exceeding LESS under token-level computation burden.** The data percentages in Table 1 are calculated at the sequence level, because other baselines are sequence-based data selection approaches. As QCS set $\alpha=0.5$, it just softly select a half of tokens in the selected sequences. In this regards, we turn to set $\frac{\alpha K}{|\mathcal{D}_{\text{train}}|}=5\%$ rather than $\frac{K}{|\mathcal{D}_{\text{train}}|}=5\%$ as another **QCS** baseline (token-level) with $\epsilon=0$ to achieve full-parameter IT, thus, with the consistent burden of token computation. The empirical study is presented in Table.1, where Pythia model series and LLAMA-2-7B are evaluated with **TYDIQA**. As observed, using small LLMs potentially expands the performance gap between **LESS** and **QCS** and more importantly, most of the cases in Pythia models fail to recover the full-data-IT performance. In particular, the performance of **QCS** exhibits a more pronounced underperformance relative to **LESS**. This observed degradation can be elucidated through the lens of Eq.(15). It implies that IT by sequence-level data selection heavily depends on the capability of base models. While interesting, under the consistent token computation, **QCS** baselines with $\frac{\alpha K}{|\mathcal{D}_{\text{train}}|}=5\%$ significantly improve the original performances: not only exceed the performance of **LESS** but also recover the LLM performance trained with full IT datasets.

## 7.2 Analysis of Token-level Selection

Here we further verify QCS ability to capture token-level information through some verification, where $\frac{K}{|\mathcal{D}_{\text{train}}|}=5\%$ is promised the token selection happen after the sequence selection.

**Elastic ratio $\epsilon$ of tokens.** We first consider how the variation of $\epsilon$ causes the vibration of target IT. In particular, given $\alpha = 0.5$, we change the value of $\epsilon$ in the range of $\{0, 0.1, 0.5, 1\}$, then testify the corresponding LLM performance trained by QCS. As observed in Fig.(2.(a)), QCS results with $\epsilon=0.1$ and $\epsilon=0.5$ are trivially distinct, whereas the performances significantly drop with a larger $\epsilon$ value. As $\epsilon=0$, the performance are inferior than $\epsilon=0.1, 0.5$.

**Selection percentage $\alpha$ of tokens.** We then turn to evaluate the change of proportion $\alpha$ about tokens selected in a single sequence completion from $\{0.2, 0.5, 1\}$ when $\epsilon = 0.1$. As illustrated in Fig.(2.(b)), the performance of QCS exhibits notable sensitivity to the value of $\alpha$. Results obtained with $\alpha$ values

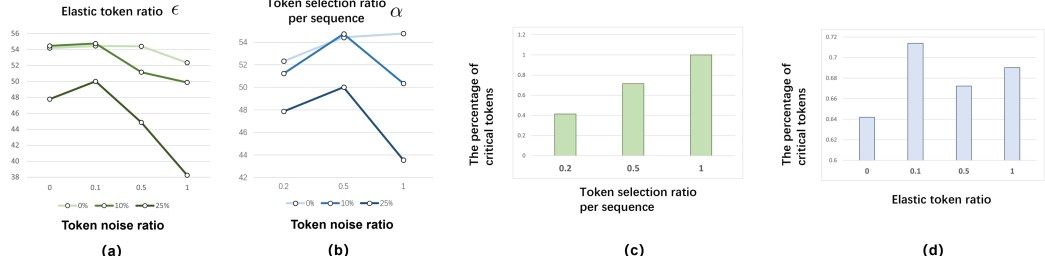

Figure 2: Parameter analysis of token-level selection.

of 0.5 and 1 demonstrate comparable efficacy, while settings where $\alpha = 0.2$ yield substantially inferior outcomes relative to other configurations.

**Token-level robustness.** We inject the random noises into the tokens [39], then further observe the performance change given the previous $\epsilon$ and $\alpha$ value. As shown in Fig.(2), we found that sufficiently large $\epsilon$ and $\alpha$ could lead to more fragile IT models, with significant performance drop with 10% token noises injection. While a small value $\epsilon$ may produce the better performance than the noiseless cases. However, as the noise ratio increases, critical tokens are more possibly contaminated, thus, all the performances drop in this scenario.

**Critical Tokens.** Critical tokens in sequences play a crucial role of efficiently following instructions with few demonstrations. In order to verify QCS's ability to select critical tokens, we consider the identification approach of critical tokens in [32], since there are no annotation of critical tokens in our evaluation benchmarks. To this, we evaluate how many tokens selected by QCS are categorized into the critical tokens identified by [32]. The results are shown in Fig.(2.(c)). As we see in Fig.(2.(c)).left, the percentage of critical tokens selected by QCS is significantly more than random selection, which demonstrates the remarkablity of QCS of sensing critical tokens while the sequence-token selection. In Fig.(2.d), we observe the variation of $\epsilon$ with tiny effect to the number of critical tokens selected.

## 7.3 Regular Efficient Instruction-Following

**Experimental Setup.** We turn to justify the QCS's feasibility in the regular data-efficient IT tasks using **LLAMA-2-7B** and **LLAMA-2-13B**. We consider the vanilla QCS algorithm to obtain the optimal $v$ and $w$ with regards to the training dataset $\mathcal{D}_{\text{train}}$ defined by the self-instruct paradigm. It is derived from two setups: **(1)** we consider the regular instruction following setup in [38] then evaluate the baselines on 3 common instruction-following benchmarks **ARC**,**TruthfulQA**, **MMLU** in Hugging-face Open LLM Leaderboard-v1[2](The evaluation results are presented in Table. in Appendix.D); **(2)** we consider **AlpaEval**[3] to evaluate LLMs trained by selecting data on FLAN V2, COT, DOLLY, and OPEN ASSISTANT 1, justifying the free-form response generation capability. In the first setup, we compared QCS with FULL, Rand, and two data-efficient IT methods, *i.e.*, e Instruction-Following Difficulty (IFD) [38] and Dataset Quantization (DQ) [69]. In the second setup, QCS is compared with those except for DQ, since it almost fails to other methods.

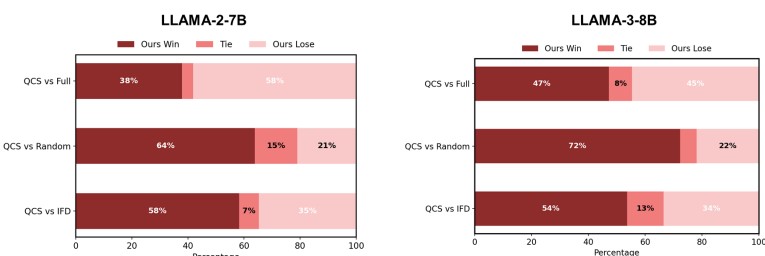

Figure 3: The AlpaEval comparison on the win rates for QCS against different other baselines.

**Results on AlpaEval.** QCS achieves a dominant win rate over Random and IDS across open-ended, instruction-following prompts. The plot shows QCS winning the large majority of pairwise

---

[2]huggingface.co/docs/leaderboards/open_llm_leaderboard

[3]github.com/tatsu-lab/alpaca_eval

Table 3: The **LLAMA-2** family trained by Data-efficient IT with (10%) instruction-response sequences, then evaluated in Huggingface Open LLM Leaderboard (**ARC**, **TruthfulQA**, **MMLU**)

| 5% | LLAMA-2-7B | | | | LLAMA-2-13B | | | |
|---|---|---|---|---|---|---|---|---|
| | Average | ARC | MMLU | TruthfulQA | Average | ARC | MMLU | TruthfulQA |
| wo sequence selection | 47.45 | 54.35 | 47.02 | 40.98 | 51.04 | 57.59 | 54.05 | 41.49 |
| DQ | 42.32 | 47.01 | 42.34 | 40.12 | 48.72 | 50.22 | 51.91 | 40.81 |
| IFD | 48.28 | 57.94 | 44.19 | 40.62 | 53.40 | 62.37 | 55.65 | 42.82 |
| QCS (token-selected 20%) | 48.86 | 58.45 | 45.54 | 41.67 | 52.43 | 61.04 | 54.46 | 42.54 |
| QCS (token-selected 50%) | **49.46** | **58.51** | **47.13** | **42.36** | **53.33** | **62.67** | **54.56** | **43.17** |

| 10% | LLAMA-2-7B | | | | LLAMA-2-13B | | | |
|---|---|---|---|---|---|---|---|---|
| | Average | ARC | MMLU | TruthfulQA | Average | ARC | MMLU | TruthfulQA |
| wo sequence selection | 47.45 | 54.35 | **47.02** | 40.98 | 51.04 | 57.59 | 54.05 | 41.49 |
| DQ | 44.38 | 48.17 | 44.55 | 40.42 | 48.72 | 52.07 | 52.89 | 41.22 |
| IFD | 48.28 | 58,02 | 46.64 | 40.18 | 53.40 | 62.97 | 55.29 | 41.93 |
| QCS (token-selected 20%) | 48.86 | 58.78 | 46.34 | 41.47 | 53.34 | 62.57 | 55.20 | 42.24 |
| QCS (token-selected 50%) | **49.46** | **59.60** | 46.53 | **42.26** | **54.03** | **63.16** | **55.86** | **43.07** |

| 15% | LLAMA-2-7B | | | | LLAMA-2-13B | | | |
|---|---|---|---|---|---|---|---|---|
| | Average | ARC | MMLU | TruthfulQA | Average | ARC | MMLU | TruthfulQA |
| wo sequence selection | 47.45 | 54.35 | **47.02** | 40.98 | 51.04 | 57.59 | 54.05 | 41.49 |
| DQ | 43.88 | 48.27 | 44.79 | 40.01 | 48.22 | 51.12 | 52.29 | 41.30 |
| IFD | 48.28 | 57.42 | 46.40 | 40.95 | 53.40 | 62.37 | 55.56 | 43.42 |
| QCS (token-selected 20%) | 48.90 | 58.58 | 46.61 | 41.87 | 53.04 | 62.23 | 55.25 | 41.81 |
| QCS (token-selected 50%) | **49.16** | **59.51** | 46.93 | **42.56** | **54.33** | **61.62** | **55.68** | **43.19** |

comparisons (e.g., 511 wins vs. 168 losses against Random), indicating that its bi-level selection reliably surfaces higher-value examples for general instruction following. While it does not surpass the model trained on the full dataset, this is expected; the key finding is that with only a small selected subset, QCS matches or approaches the full-data model and clearly beats other selection heuristics. The advantage is most evident in long-form responses, where token-level prioritization highlights factual terms and connective logic, yielding more informative updates than sequence-only curation. Consistency across both modern (LLAMA-3) and established (LLAMA-2) bases suggests QCS is robust to the backbone's capability and provides a principled, data-efficient alternative for regular IT.

**Results on ARC,TruthfulQA, MMLU**. Distinct from ApacEval, the experimental evaluation also includes two QCS baselines by ranging sequence-selection ratio is ranged from 5%, 10%, 15%, respectively; and our QCS is implemented with 20%, 50% token ratios to derive two variants. The results are observed in Table.3. Under the regular efficient IT setup, QCS with token selection (especially 50%) delivers the best or near-best averages on LLAMA-2-7B/13B, outperforming IFD and DQ and often exceeding "no selection" at the same budget. Gains come from emphasizing salient tokens within selected sequences, not just choosing sequences. However, improvements are smaller than on AlpacaEval: these benchmarks mostly require short outputs (single-token or brief spans), limiting token-level leverage; hence QCS's benefit over strong sequence-only methods is modest but consistent. Overall, QCS remains competitive across all three leaderboards and confirms that its largest margin emerges on tasks with longer, content-rich responses.

# 8 Conclusion

We introduced Quadratic Coreset Selection (QCS), a principled framework that reconciles sequence-level and token-level mining for data-efficient instruction tuning. By casting both selections into a unified bi-level coreset objective and relaxing it via a probabilistic re-parameterization, QCS avoids Hessian inverses, learns with hierarchical policy gradients, and enjoys provable convergence with asymptotic equivalence to the original objective. Empirically, QCS delivers superior or competitive performance under stringent data budgets, particularly on long-form, reasoning-heavy tasks, and adapts well to targeted and continual IT. Analyses reveal token-level benefits—robustness to noise, emphasis on critical tokens, and sensitivity to elastic weighting—clarifying when and why QCS outperforms sequence-only selection. QCS thus offers a scalable, theoretically grounded path to efficient post-training.

## Acknowledgement

The research was supported in part by Guangdong S&T Programme (Grant No. 2024B0101010003); in part by Open research fund of Pengcheng Laboratory under Grant 2025KF1B0050, in part by The Major Key Project of PCL (No. PCL2024A04, PCL2025A02); in part by National Natural Science Foundation of China (NSFC) under Grant No.62206110, 62176103, 62377208,62572498, and 62276114; in part by the Science and Technology Planning Project of Guangzhou under grants 2024A04J9896, 2025A03J3565.National Natural Science Foundation of China under Grant No. 62572498

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

## Appendix.A: Technical Discussions

In this section, we unveil the connection between QCS and empirical influence functions, and provide the back-propagation free solution of **w** and **v**.

### The Interpretability of QCS via Empirical Influence

To be selected by the original QCS objective (5), the token in the position $(i,j)$, *i.e.*. at the $j$-th order of the $i$-th sequence, have to satisfy the following two condition: (1). provided the optimal $\boldsymbol{w}^* \in \mathcal{C}_{\text{seq}}$, $w_i^* = 1$; (2). provided the optimal $\boldsymbol{v}^* \in \mathcal{C}_{\text{tkn}}$, $v_{i,j}^* = 1$. The solutions are derived from the formulations

$$\min_{\boldsymbol{w} \in \mathcal{C}_{\text{seq}}} \mathcal{L}(\theta^*(\boldsymbol{w}, \boldsymbol{v})) = \frac{1}{M} \sum_{i=1}^{M} \Big( \frac{1}{N_i} \sum_{i=1}^{N_i} \mathcal{L}_{i,j}(\theta^*(\boldsymbol{w}, \boldsymbol{v})) \Big), \text{s.t.}, \theta^*(\boldsymbol{v}) \in \arg\min_{\theta \in \Theta} \hat{\mathcal{L}}(\theta(\boldsymbol{w}, \boldsymbol{v})),$$

$$\hat{\mathcal{L}}(\theta(\boldsymbol{w}, \boldsymbol{v})) = \frac{1}{K} \sum_{i=1}^{K} \Big( \frac{1}{K_i} \sum_{j=1}^{N_i} v_{i,j} \mathcal{L}_{i,j}(\theta(\boldsymbol{w}, \boldsymbol{v})) \Big) \tag{16}$$

and

$$\min_{\boldsymbol{v} \in \mathcal{C}_{\text{tkn}}} \mathcal{L}(\theta^*(\boldsymbol{w}, \boldsymbol{v})) = \frac{1}{M} \sum_{i=1}^{M} \Big( \frac{1}{N_i} \sum_{i=1}^{N_i} \mathcal{L}_{i,j}(\theta^*(\boldsymbol{w}, \boldsymbol{v})) \Big), \text{s.t.}, \theta^*(\boldsymbol{v}) \in \arg\min_{\theta \in \Theta} \hat{\mathcal{L}}(\theta(\boldsymbol{w}, \boldsymbol{v})),$$

$$\hat{\mathcal{L}}(\theta(\boldsymbol{w}, \boldsymbol{v})) = \frac{1}{K} \sum_{i=1}^{K} \Big( \frac{1}{K_i} \sum_{j=1}^{N_i} v_{i,j} \mathcal{L}_{i,j}(\theta(\boldsymbol{w}, \boldsymbol{v})) \Big). \tag{17}$$

According to the approximation results derived from cone constrained optimization [6], we got the greedy search algorithm with the selection rule

$$\text{token } (i^*, j^*) \in \Big\{ j^* : j^* \in \arg\max_{j \in [N_{i^*}]} \nabla_\theta w_{i^*} \mathcal{L}_{i^*,j}^\top \Big( \frac{\partial f(\theta(\boldsymbol{w}_{S_{\text{seq}}^{t_1}}, \boldsymbol{v}_{S_{\text{tkn}}^{t_2}}))}{\partial \theta^\top \theta} \Big)^{-1} \big(\nabla_\theta(\mathcal{L}(\theta))\big) \Big\},$$

$$\text{s.t. } w_{i^*} \in \boldsymbol{w}_{S_{\text{seq}}^{t_1}},$$

$$\leftarrow \text{token } (i^*, j^*) \in \Big\{ j^* : j^* \in \arg\max_{j \in [N_i]} \nabla_\theta \mathcal{L}_{i,j}^\top \Big( \frac{\partial f(\theta(\boldsymbol{w}_{S_{\text{seq}}^{t_1}}, \boldsymbol{v}_{S_{\text{tkn}}^{t_2}}))}{\partial \theta^\top \theta} \Big)^{-1} \big(\nabla_\theta(\mathcal{L}(\theta))\big) \Big\}, \tag{18}$$

$$\text{s.t. } i \in \Big\{ i^* : i^* \in \arg\max_{i \in [N]} \Big( \frac{1}{N_i} \sum_{k \in [N_i]} \nabla_\theta v_{i,k} \mathcal{L}_{i,k} \Big)^\top \Big( \frac{\partial f(\theta(\boldsymbol{w}_{S_{\text{seq}}^{t_1}}, \boldsymbol{v}_{S_{\text{tkn}}^{t_2}}))}{\partial \theta^\top \theta} \Big)^{-1} \big(\nabla_\theta(\mathcal{L}(\theta))\big) \Big\},$$

$$\text{w.r.t. } v_{i,k} \in \boldsymbol{v}_{S_{\text{tkn}}^{t_1}},$$

where $S_{\text{seq}}^{t_1}$, $S_{\text{tkn}}^{t_2}$ denote a set of sequence atoms of size $t_1$ ($t_1 \leq K$) and a set of token atoms of size $t_2 (t_2 \leq K_i)$ have been selected, respectively. Note that, A.(18) is closely related with the token-level empirical influence $\frac{\partial \theta^*}{\partial \varepsilon_{\text{tkn}}}\Big|_{\varepsilon_{\text{tkn}}=0} = -\Big( \frac{\partial f(\theta(\boldsymbol{w}^*, \boldsymbol{v}^*))}{\partial \theta^\top \theta} \Big)^{-1} \nabla_\theta \mathcal{L}_{i,j}(\theta^*)$ and the sequence-level empirical influence $\frac{\partial \theta^*}{\partial \varepsilon_{\text{seq}}}\Big|_{\varepsilon_{\text{seq}}=0} = -\Big( \frac{\partial f(\theta(\boldsymbol{w}^*, \boldsymbol{v}^*))}{\partial \theta^\top \theta} \Big)^{-1} \Big( \frac{1}{N_i} \sum_{k=1}^{N_i} \nabla_\theta \mathcal{L}_{i,k}(\theta^*) \Big)$. In particular, if the greedy search results $\boldsymbol{w}_{S_{\text{seq}}^{t_1}} = \boldsymbol{w}^*$ and $\boldsymbol{v}_{S_{\text{tkn}}^{t_2}} = \boldsymbol{v}^*$, A.(18) holds

$$\text{token } (i^*, j^*) \in \Big\{ j^* : j^* \in \arg\max_{j \in [N_i]} -\Big( \frac{\partial \theta^*}{\partial \varepsilon_{\text{tkn}}}\Big|_{\varepsilon_{\text{tkn}}=0} \Big)^\top \big(\nabla_\theta(\mathcal{L}(\theta^*))\big) \Big\},$$

$$\text{s.t. } i \in \Big\{ i^* : i^* \in \arg\max_{i \in [N]} -\Big( \frac{\partial \theta^*}{\partial \varepsilon_{\text{seq}}}\Big|_{\varepsilon_{\text{seq}}=0} \Sigma(\boldsymbol{v}^*) \Big)^\top \big(\nabla_\theta(\mathcal{L}(\theta^*))\big) \Big\},$$

$$\text{w.r.t. } \Sigma_{m,n}(\boldsymbol{v}) = 0 \ (m \neq n); \quad \Sigma_{m,n}(\boldsymbol{v}) = \frac{\big( \frac{1}{N_i} \sum_{k \in [N_i]} \nabla_{\theta_m} v_{i,k} \mathcal{L}_{i,k} \big)^\top}{\big( \frac{1}{N_i} \sum_{k \in [N_i]} \nabla_{\theta_m} \mathcal{L}_{i,k} \big)^\top} \ (m = n) \tag{19}$$

where $\nabla_{\theta_m} \mathcal{L}$ indicates the $m$-th element of vector $\nabla_\theta \mathcal{L}$.

**The Coreset Sampler Update without Back-propagation**

We consider the projection $\mathbf{w} \leftarrow \mathcal{P}_{\mathcal{C}_{\text{seq}}}\big(\mathbf{w} - \eta_1 \mathcal{L}(\theta^*(\boldsymbol{w}, \boldsymbol{v}))\big)\nabla_{\mathbf{w}} \ln p(\boldsymbol{w}|\mathbf{w})$, then $\mathbf{v} \leftarrow \mathcal{P}_{\mathcal{C}_{\text{tkn}}}\big(\mathbf{v} - \eta_2 \mathcal{L}(\theta^*(\boldsymbol{w}, \boldsymbol{v}))\big)\nabla_{\mathbf{v}} \ln p(\boldsymbol{v}|\boldsymbol{w} \odot \mathbf{v})$ can be solved accordingly. Specifically, we assume the projected result is $\mathbf{w}$, then the origin denotes as $\mathbf{z}_{\mathbf{w}}$, to solve $\mathbf{w} \leftarrow \mathcal{P}_{\mathcal{C}_{\text{seq}}}\big(\mathbf{w} - \eta_1 \mathcal{L}(\theta^*(\boldsymbol{w}, \boldsymbol{v}))\big)\nabla_{\mathbf{w}} \ln p(\boldsymbol{w}|\mathbf{w})$ refers to

$$\min_{\mathbf{w} \in \mathbb{R}^N} ||\mathbf{w} - \mathbf{z}_{\mathbf{w}}||_2, \text{ s.t. } \mathbf{1}^\top \mathbf{w} \leq K, \ 0 \leq \mathrm{w}_i \leq 1. \tag{20}$$

It refers to solving the problem with Lagrangian multipliers

$$\min_{\mathbf{w} \in \mathbb{R}^N} ||\mathbf{w} - \mathbf{z}_{\mathbf{w}}||^2 + \lambda_1(\mathbf{1}^\top \mathbf{w} - K)$$
$$= \min_{\mathbf{w} \in \mathbb{R}^N} ||\mathbf{w} - (\mathbf{z}_{\mathbf{w}} - \lambda \mathbf{1})||^2 + \lambda_1(\mathbf{1}^\top \mathbf{w} - K) - \frac{N}{2}\lambda^2, \text{ s.t. } \lambda > 0, \ 0 \leq \mathrm{w}_i \leq 1. \tag{21}$$

To minimize the formulation with respect to $\boldsymbol{w}$, we have

$$\mathbf{w} = \mathbf{1}_{\mathbf{z}_{\mathbf{w}} - \lambda\mathbf{1} \geq 1} + (\mathbf{z}_{\mathbf{w}} - \lambda\mathbf{1})_{1 > \mathbf{z}_{\mathbf{w}} - \lambda\mathbf{1} > 0}. \tag{22}$$

So given $\lambda \geq 0$, we have

$$g(\lambda) = \frac{1}{2}||[\mathbf{z}_{\mathbf{w}} - \lambda\mathbf{1}]_-||^2 + \frac{1}{2}||[\mathbf{z}_{\mathbf{w}} - (\lambda+1)\mathbf{1}]_+||^2 + \lambda(\mathbf{1}^\top \mathbf{z}_{\mathbf{w}} - \mathbf{w}) - \frac{N}{2}\lambda^2$$
$$\rightarrow g'(\lambda) = \mathbf{1}^\top \min(1, \max(0, \mathbf{z}_{\mathbf{w}} - \lambda\mathbf{1})) - K \tag{23}$$

Since $g'(\lambda)$ is monotone decreasing with respect to $\lambda$ so that we can solve $g'(\lambda) = 0$ with solution $\lambda_1^*$ where the maximum of $g(\lambda)$ is achieved at 0 if $\lambda_1^* \leq 0$ and $\lambda_1^*$ if $\lambda_1^* > 0$. Set $\lambda^* = \max\{0, \lambda_1^*\}$, it holds

$$\mathbf{w}^* = \mathbf{1}_{\mathbf{z}_{\mathbf{w}} - \max\{0,\lambda_1^*\}\mathbf{1} \geq 1} + (\mathbf{z}_{\mathbf{w}} - \max\{0,\lambda_1^*\}\mathbf{1})_{1 > \mathbf{z}_{\mathbf{w}} - \max\{0,\lambda_1^*\}\mathbf{1} > 0}$$
$$= \mathbf{1}^\top \min(1, \max(0, \mathbf{z}_{\mathbf{w}} - \max\{0, \lambda_1^*\}\mathbf{1})) \tag{24}$$

Similarly, $\forall \boldsymbol{w} \sim p(\boldsymbol{w}|\mathbf{w})$ we have

$$\mathbf{v}^* = \mathbf{1}_{\boldsymbol{w} \odot \mathbf{z}_{\mathbf{v}} - \max\{0,\lambda_2^*\}\mathbf{1} \geq 1} + (\boldsymbol{w} \odot \mathbf{z}_{\mathbf{v}} - \max\{0,\lambda_2^*\}\mathbf{1})_{1 > \boldsymbol{w} \odot \mathbf{z}_{\mathbf{v}} - \max\{0,\lambda_2^*\}\mathbf{1} > 0}$$
$$= \mathbf{1}^\top \min(1, \max(0, \boldsymbol{w} \odot \mathbf{z}_{\mathbf{v}} - \max\{0, \lambda_2^*\}\mathbf{1})) \tag{25}$$

where $\lambda_2^* \geq 0$ indicates the maximum of $h(\lambda)$:

$$h(\lambda) = \frac{1}{2}||[\boldsymbol{w} \odot \mathbf{z}_{\mathbf{v}} - \lambda\mathbf{1}]_-||^2 + \frac{1}{2}||[\boldsymbol{w} \odot \mathbf{z}_{\mathbf{v}} - (\lambda+1)\mathbf{1}]_+||^2 + \lambda(\mathbf{1}^\top \boldsymbol{w} \odot \mathbf{z}_{\mathbf{v}} - \mathbf{v}) - \frac{N}{2}\lambda^2$$
$$\rightarrow h'(\lambda) = \mathbf{1}^\top \min(1, \max(0, \boldsymbol{w} \odot \mathbf{z}_{\mathbf{v}} - \lambda\mathbf{1})) - \sum_{i=1}^{K} K_i \tag{26}$$

## Appendix.B: Theoretical Justification

In this section, we provide the proofs of our theoretical results in the paper.

**Proof of Theorem.1**

*Proof.* Let $S_{\text{seq}}^{(\phi,K)}(\boldsymbol{w}) := \Big\{\theta|\theta(\boldsymbol{w}) \in \arg\min_{\theta' \in \Theta} \frac{1}{K}\sum_{i=1}^{M} w_i\big(\frac{1}{N_i}\sum_{j=1}^{N_i} \mathcal{L}_{i,j}(\theta')\big), \text{s.t., } \boldsymbol{w} \in \{0,1\}^M, |\boldsymbol{w}| \leq K\Big\}$ and $S_{\text{tkn}}^{(\phi,\{K_i\}_{i=1}^M)}(\boldsymbol{v}) := \Big\{\theta|\theta(\boldsymbol{v}) \in \arg\min_{\theta' \in \Theta} \frac{1}{M}\sum_{i=1}^{M}\big(\frac{1}{K_i}\sum_{j=1}^{N_i} v_{i,j}\mathcal{L}_{i,j}(\theta')\big), \text{s.t., } \boldsymbol{v}_i \in \{0,1\}^{N_i}, |\boldsymbol{v}_i| \leq K_i, \forall i \in [M]\Big\}$. Given this, The solution sets of (3) and (4) can be rewritten as

$$\mathcal{S}_{\text{seq}} = \Big\{(\boldsymbol{w},\theta)|(\boldsymbol{w},\theta) \in \arg\min_{\boldsymbol{w} \in \mathcal{C}_{\text{seq}}, \theta \in S_{\text{seq}}^{(\phi,K)}(\boldsymbol{w})} \frac{1}{M}\sum_{i=1}^{M}\big(\frac{1}{N_i}\sum_{i=1}^{N_i} \mathcal{L}_{i,j}(\theta)\big)\Big\};$$
$$\mathcal{S}_{\text{tkn}} = \Big\{(\boldsymbol{v},\theta)|(\boldsymbol{v},\theta) \in \arg\min_{\boldsymbol{v} \in \mathcal{C}_{\text{tkn}}, \theta \in S_{\text{tkn}}^{(\phi,\{K_i\}_{i=1}^M)}(\boldsymbol{v})} \frac{1}{M}\sum_{i=1}^{M}\big(\frac{1}{N_i}\sum_{i=1}^{N_i} \mathcal{L}_{i,j}(\theta)\big)\Big\}. \tag{27}$$

Let $S_{\text{qcs}}^{(\phi)}(\boldsymbol{v}, \boldsymbol{w}) := \Big\{ \theta | \theta(\boldsymbol{w}, \boldsymbol{v}) \in \arg\min_{\theta' \in \Theta} \frac{1}{K} \sum_{i=1}^{M} w_i \big( \frac{1}{N_i} \sum_{j=1}^{N_i} v_{i,j} \mathcal{L}_{i,j}(\theta') \big), \text{s.t. } \boldsymbol{w} \in \{0,1\}^M, |\boldsymbol{w}| \leq K; \boldsymbol{v}_i \in \{0,1\}^{N_i}, |\boldsymbol{v}_i| \leq K_i, \forall i \in [M] \Big\}$. The solution set of QCS can be rewritten into

$$\mathcal{S}_{\text{qcs}} = \Big\{ (\boldsymbol{w}, \boldsymbol{v}, \theta) | (\boldsymbol{w}, \boldsymbol{v}, \theta) \in \arg\min_{\boldsymbol{w} \in \mathcal{C}_{\text{seq}}, \boldsymbol{v} \in \mathcal{C}_{\text{tkn}}, \theta \in S_{\text{qcs}}^{(\phi)}(\boldsymbol{w}, \boldsymbol{v})} \frac{1}{M} \sum_{i=1}^{M} \big( \frac{1}{N_i} \sum_{i=1}^{N_i} \mathcal{L}_{i,j}(\theta) \big) \Big\}. \quad (28)$$

Observe that if we set $K = M$ and $\boldsymbol{w} = \boldsymbol{1}^M$, the equivalence holds

$$S_{\text{qcs}}^{(\phi)}(\boldsymbol{1}^M, \boldsymbol{v})$$

$$= \Big\{ \theta | \theta(\boldsymbol{1}^M, \boldsymbol{v}) \in \arg\min_{\theta' \in \Theta} \frac{1}{K} \sum_{i=1}^{M} \big( \frac{1}{N_i} \sum_{j=1}^{N_i} v_{i,j} \mathcal{L}_{i,j}(\theta') \big), \text{s.t. } \boldsymbol{v}_i \in \{0,1\}^{N_i}, |\boldsymbol{v}_i| \leq K_i, \forall i \in [M] \Big\}$$

$$= S_{\text{tkn}}^{(\phi, \{K_i\}_{i=1}^M)}(\boldsymbol{v}).$$
$$(29)$$

Similarly, if $\forall i \in [M]$ w.r.t. $w_i = 1$, $v_{i,j} = 1$, $j \in [N_i]$. We set $\hat{\boldsymbol{v}} = \{\boldsymbol{v} | \boldsymbol{v}_i = \boldsymbol{1}^M \text{ iff } w_i = 1\}$, then

$$S_{\text{qcs}}^{(\phi)}(\boldsymbol{w}, \hat{\boldsymbol{v}})$$

$$= \Big\{ \theta | \theta(\boldsymbol{w}, \hat{\boldsymbol{v}}) \in \arg\min_{\theta' \in \Theta} \frac{1}{K} \sum_{i=1}^{M} w_i \big( \frac{1}{N_i} \sum_{j=1}^{N_i} \mathcal{L}_{i,j}(\theta') \big), \text{s.t., } \boldsymbol{w} \in \{0,1\}^M, |\boldsymbol{w}| \leq K \Big\} \quad (30)$$

$$= S_{\text{seq}}^{(\phi, K)}(\boldsymbol{w}).$$

Therefore, we have

$$\mathcal{S}_{\text{seq}} = \Big\{ (\boldsymbol{w}, \theta) | (\boldsymbol{w}, \theta) \in \arg\min_{\boldsymbol{w} \in \mathcal{C}_{\text{seq}}, \theta \in S_{\text{seq}}^{(\phi, K)}(\boldsymbol{w})} \frac{1}{M} \sum_{i=1}^{M} \big( \frac{1}{N_i} \sum_{i=1}^{N_i} \mathcal{L}_{i,j}(\theta) \big) \Big\}$$

$$= \Big\{ (\boldsymbol{w}, \boldsymbol{v}, \theta) | (\boldsymbol{w}, \boldsymbol{v}, \theta) \in \arg\min_{\boldsymbol{w} \in \mathcal{C}_{\text{seq}}, \boldsymbol{v} = \hat{\boldsymbol{v}}, \theta \in S_{\text{seq}}^{(\phi)}(\boldsymbol{w}, \hat{\boldsymbol{v}})} \frac{1}{M} \sum_{i=1}^{M} \big( \frac{1}{N_i} \sum_{i=1}^{N_i} \mathcal{L}_{i,j}(\theta) \big) \Big\}$$
$$(31)$$

and

$$\mathcal{S}_{\text{tkn}} = \Big\{ (\boldsymbol{v}, \theta) | (\boldsymbol{v}, \theta) \in \arg\min_{\boldsymbol{v} \in \mathcal{C}_{\text{tkn}}, \theta \in S_{\text{tkn}}^{(\phi, \{K_i\}_{i=1}^M)}(\boldsymbol{v})} \frac{1}{M} \sum_{i=1}^{M} \big( \frac{1}{N_i} \sum_{i=1}^{N_i} \mathcal{L}_{i,j}(\theta) \big) \Big\}$$

$$= \Big\{ (\boldsymbol{1}^M, \boldsymbol{v}, \theta) | (\boldsymbol{1}^M, \boldsymbol{v}, \theta) \in \arg\min_{\boldsymbol{w} = \boldsymbol{1}^M, \boldsymbol{v} \in \mathcal{C}_{\text{tkn}}, \theta \in S_{\text{qcs}}^{(\phi)}(\boldsymbol{1}^M, \boldsymbol{v})} \frac{1}{M} \sum_{i=1}^{M} \big( \frac{1}{N_i} \sum_{i=1}^{N_i} \mathcal{L}_{i,j}(\theta) \big) \Big\}.$$
$$(32)$$

Observe that $\hat{\boldsymbol{v}} \subset \mathcal{C}_{\text{tkn}}$ and $S_{\text{seq}}^{(\phi)}(\boldsymbol{w}, \hat{\boldsymbol{v}}) \subset S_{\text{seq}}^{(\phi)}(\boldsymbol{w}, \boldsymbol{v})$. $\forall \theta_{\text{qcs}}^* \in \Omega(\theta_{\text{qcs}}) = \{\theta | (\boldsymbol{w}, \boldsymbol{v}, \theta) \in \mathcal{S}_{\text{qcs}}\}$ and $\forall \theta_{\text{tkn}}^* \in \Omega(\theta_{\text{tkn}}) = \{\theta | (\boldsymbol{w}, \theta) \in \mathcal{S}_{\text{tkn}}\}$, it holds

$$\frac{1}{M} \sum_{i=1}^{M} \big( \frac{1}{N_i} \sum_{i=1}^{N_i} \mathcal{L}_{i,j}(\theta_{\text{qcs}}^*) \big) \leq \frac{1}{M} \sum_{i=1}^{M} \big( \frac{1}{N_i} \sum_{i=1}^{N_i} \mathcal{L}_{i,j}(\theta_{\text{tkn}}^*) \big). \quad (33)$$

It implies $\forall \theta_{\text{qcs}}^* \in \Omega(\theta_{\text{qcs}})$, $\theta_{\text{qcs}}^* \in \Omega(\theta_{\text{tkn}})$. Since $\{\boldsymbol{1}^M\} \subset \mathcal{C}_{\text{seq}}$ and $S_{\text{seq}}^{(\phi)}(\boldsymbol{1}^M, \boldsymbol{v}) \subset S_{\text{seq}}^{(\phi)}(\boldsymbol{w}, \boldsymbol{v})$, $\forall \theta_{\text{qcs}}^* \in \Omega(\theta_{\text{qcs}}) = \{\theta | (\boldsymbol{w}, \boldsymbol{v}, \theta) \in \mathcal{S}_{\text{qcs}}\}$ and $\forall \theta_{\text{seq}}^* \in \Omega(\theta_{\text{seq}}) = \{\theta | (\boldsymbol{w}, \theta) \in \mathcal{S}_{\text{seq}}\}$ also result in

$$\frac{1}{M} \sum_{i=1}^{M} \big( \frac{1}{N_i} \sum_{i=1}^{N_i} \mathcal{L}_{i,j}(\theta_{\text{qcs}}^*) \big) \leq \frac{1}{M} \sum_{i=1}^{M} \big( \frac{1}{N_i} \sum_{i=1}^{N_i} \mathcal{L}_{i,j}(\theta_{\text{seq}}^*) \big). \quad (34)$$

It implies $\forall \theta_{\text{qcs}}^* \in \Omega(\theta_{\text{qcs}})$, $\theta_{\text{qcs}}^* \in \Omega(\theta_{\text{seq}})$. Summarize (33,34) and we have $\Omega(\theta_{\text{qcs}}) \subset \Omega(\theta_{\text{seq}}) \cap \Omega(\theta_{\text{tkn}})$. $\qquad\square$

**Proofs of Theorem.2**

There are some assumptions required to justify the asymptotic approximation between the original QCS and its probabilistic variant (Theorem.2).

**Assumption 4.** (**(Lipschitz Continuity)**) The next-token prediction loss $\mathcal{L}_{i,j}$ is $L$-Lipschitz with respect to the selection weights, thus, $\forall \mathbf{v}, \mathbf{v}' \in \mathcal{C}_{\mathsf{tkn}}$ and $\mathbf{w}, \mathbf{w}' \in \mathcal{C}_{\mathsf{seq}}$, $||\mathcal{L}_{i,j}(\theta(\mathbf{w}, \mathbf{v})) - \mathcal{L}_{i,j}(\theta(\mathbf{w}', \mathbf{v}'))|| \leq L(||\mathbf{w} - \mathbf{w}'|| + ||\mathbf{v} - \mathbf{v}'||)$

**Assumption 5.** (**Tokenwise Boundedness**) The per-token loss $\mathcal{L}_{i,j}(\cdot)$ are uniformly bounded (or, more generally, sub-Gaussian).

**Assumption 6.** (**Large Sample Size**) We assume that the *effective* number of independent selections, denoted by $n$ (which could be the total number of tokens or sequences), tends to infinity.

**The validity of assumptions.** In general, the boundedness and Lipschitz continuity conditions for the next-token prediction loss with respect to the selection weights serves for two conditions, thus, the loss serves Lipschitz continuity to the model parameter; and the model parameter satisfies the Lipschitz continuity to the selection weights. The first case is practical. Specifically, the tokenwise prediction loss $\mathcal{L}_{i,j} = -\log P(x_{i,j} | \boldsymbol{x}_{i,<j}, \boldsymbol{z}_i; \theta)$ is often defined as the cross-entropy loss or negative log-likelihood so that $\mathcal{L}_{i,j} \in [0, +\infty)$. though in practical implementations, $\mathcal{L}_{i,j}$ is commonly capped due to numerical stability (*e.g.*, $\mathcal{L}_{i,j} \leq -\log \epsilon$ where $\epsilon > 0$ is the smallest representable probability), in order to avoid issues with extremely small probabilities. In the second condition, thus, the model parameter $\theta$ satisfies the Lipschitz continuity to the selection weights $\mathbf{v}$ and $\mathbf{w}$. Observe that the gradients of $\Phi$ with respect to $\mathbf{v}$ and $\mathbf{w}$ is bounded with respect to the policy gradients in (11,12), and the model parameter holds the Lipschitz continuity to $\Phi$ in terms of its the composite of $\mathcal{L}i, j$ in probability. So the second condition also holds.

$\mathcal{L}_{i,j}$ is also tokenwise bounded due to the smoothness of the softmax function (the token-prediction head) and bounded gradients with respect to the model parameters. Assumption.3 implies the deviations induced by the Bernoulli sampling can be controlled via concentration inequalities. Note that it actually holds since given the optimized $\mathbf{v}$, the selection of $v_{i,j}$ in each token position is totally random, so do $\mathbf{w}$.

*Proof.* Consider $\mathbf{w}$ and $\mathbf{v}$ obtained by optimization. Let $(\boldsymbol{w}^s, \boldsymbol{v}^s)$ be one realization of the binary selections so that

$$\boldsymbol{w}^s \sim p(\boldsymbol{w}|\mathbf{w}), \boldsymbol{v}^s \sim p(\boldsymbol{v}|\mathbf{w} \odot \mathbf{v}) \tag{35}$$

Since the probabilistic objective is the expectation of $\mathcal{L}(\theta(\boldsymbol{w}, \boldsymbol{v}))$ with $\boldsymbol{w}, \boldsymbol{v}$ drawn from the optimal $\mathbf{w}$ and $\mathbf{v}$, thus

$$\Phi(\mathbf{w}, \mathbf{v}) = \mathbb{E}_{p(\boldsymbol{w}|\mathbf{w})p(\boldsymbol{v}|\mathbf{w}\odot\mathbf{v})} \mathcal{L}(\theta^*(\boldsymbol{w}^s, \boldsymbol{v}^s)). \tag{36}$$

By construction, $\mathbf{w}$ and $\mathbf{v}$ hold $\mathbf{w}_i = \mathbb{E}_{p(\boldsymbol{w}|\mathbf{w})p(\boldsymbol{v}|\mathbf{w}\odot\mathbf{v})}[\boldsymbol{w}_i^s]$ and $\mathbf{v}_{i,j} = \mathbb{E}_{p(\boldsymbol{w}|\mathbf{w})p(\boldsymbol{v}|\mathbf{w}\odot\mathbf{v})}[\boldsymbol{v}_{i,j}^s]$ due to the definition of Bernoulli distributions. It is observed that the objective with respect to $\mathbf{w}$ and $\mathbf{v}$ satisfy the Lipschitz continuity, thus, we have for any realization

$$\begin{aligned}
&||\mathcal{L}(\theta^*(\boldsymbol{w}^s, \boldsymbol{v}^s)) - \mathcal{L}(\theta^*(\mathbf{w}, \mathbf{v}))|| \\
&= \lim_{\mathbf{w}' \to \boldsymbol{w}^s, \mathbf{v}' \to \boldsymbol{v}^s} ||\mathcal{L}(\theta^*(\mathbf{w}', \mathbf{v}')) - \mathcal{L}(\theta^*(\mathbf{w}, \mathbf{v}))|| \\
&\leq \lim_{\mathbf{w}' \to \boldsymbol{w}^s, \mathbf{v}' \to \boldsymbol{v}^s} L(||\mathbf{v} - \mathbf{v}'|| + ||\mathbf{w}' - \mathbf{w}||) \\
&= L(||\mathbf{v} - \boldsymbol{v}^s|| + ||\boldsymbol{w}^s - \mathbf{w}||)
\end{aligned} \tag{37}$$

Since the deviations $||\mathbf{w} - \boldsymbol{w}^s||$ and $||\mathbf{v} - \boldsymbol{v}^s||$ are sums of independent bounded (or sub-Gaussian) random variables, the central limit theorem applies. More specifically, $\boldsymbol{w}_i^s$ and $\boldsymbol{v}_{i,j}^s$ are binary and lie in $\{0, 1\}$, their differences from the continuous weights are bounded. By applying Hoeffding's inequality to these independent binary variables, we can assert that with probability at least $1 - \delta$

$$||\mathbf{v} - \boldsymbol{v}|| + ||\boldsymbol{w} - \mathbf{w}|| \leq C\sqrt{\frac{\log(1/\delta)}{n}} \tag{38}$$

for some constant $C > 0$ and effective number of independent samples $n$. It allows that

$$||\mathcal{L}(\theta^*(\boldsymbol{w}^s, \boldsymbol{v}^s)) - \mathcal{L}(\theta^*(\mathbf{w}, \mathbf{v}))|| \leq LC\sqrt{\frac{\log(1/\delta)}{n}} \tag{39}$$

Taking the expectation over the binary selections yields

$$||\mathcal{L}(\theta^*(\boldsymbol{w}^s, \boldsymbol{v}^s)) - \Phi(\mathbf{w}, \mathbf{v})|| \leq O(L\sqrt{\frac{\log(1/\delta)}{n}}) \tag{40}$$

$\square$

**Proof of Proposition.3**

Let first provide the formal convergence result derived from

**Proposition 7.** *Suppose that* $\Phi(\mathbf{w}, \mathbf{v})$ *is L-smooth with respect to* **w**,**v**, *respectively; and the policy gradient variance* $\mathbb{E}\big(\mathcal{L}(\theta^*(\boldsymbol{w}, \boldsymbol{v}))\nabla_{\mathbf{w}} \ln p(\boldsymbol{w}|\mathbf{w}^t) - \nabla_{\mathbf{w}}\Phi(\mathbf{w}^t, \mathbf{v}^t)\big) \leq \sigma^2$, *and* $\mathbb{E}\big(\mathcal{L}(\theta^*(\boldsymbol{w}, \boldsymbol{v}))\nabla_{\mathbf{v}} \ln p(\boldsymbol{v}|\boldsymbol{w} \odot \mathbf{v}^t) - \nabla_{\mathbf{v}}\Phi(\mathbf{w}^t, \mathbf{v}^t)\big) \leq \sigma^2$. *Let* $\eta_1, \eta_2 \leq \frac{1}{L}$ *then it holds*

$$\frac{1}{T}\sum_{t=1}^{T}\mathbb{E}\Big(||\mathcal{G}_{\mathbf{w}}^t||^2 + ||\mathcal{G}_{\mathbf{w}}^t||^2\Big) \leq 2\sigma^2\big(2 + \frac{1}{\min_{\eta \in \{\eta_1, \eta_2\}}(1 - L\eta/2)}\big) \tag{41}$$

*where* $\mathcal{G}_{\mathbf{w}}^t$, $\mathcal{G}_{\mathbf{v}}^t$ *denote the gradient mappings with respect to* **w**, **v** *at the* $t$-*th iteration,* i.e.

$$\mathcal{G}_{\mathbf{w}}^t = \frac{1}{\eta_1}\big(\mathbf{w}_t - \mathcal{P}_{\mathcal{C}_{\mathrm{seq}}}\big(\mathbf{w}^t - \eta_1\nabla_{\mathbf{w}}\Phi(\mathbf{w}^t, \mathbf{v}^t)\big)\big), \quad \mathcal{G}_{\mathbf{v}}^t = \frac{1}{\eta_2}\big(\mathbf{v}_t - \mathcal{P}_{\mathcal{C}_{\mathrm{tkn}}}\big(\mathbf{v}^t - \eta_2\nabla_{\mathbf{v}}\Phi(\mathbf{w}^t, \mathbf{v}^t)\big)\big). \tag{42}$$

Note that Proposition.3 can be proved when the proposition above is satisfied. To prove the proposition above, we first present the lemmas derived from [23]:

**Lemma 8.** *Given a compact convex set* $\mathcal{C} \subset \mathbb{R}^d$ *and let* $\mathcal{P}_\mathcal{C}$ *be the projection operator on* $\mathcal{C}$, *then for any* $u, v \in \mathbb{R}^d$, *we have*

$$||\mathcal{P}_\mathcal{C}(v) - \mathcal{P}_\mathcal{C}(u)|| \leq (v - u)^\top\big(\mathcal{P}_\mathcal{C}(v) - \mathcal{P}_\mathcal{C}(u)\big) \tag{43}$$

**Lemma 9.** *Given a compact convex set* $\mathcal{C} \subset \mathbb{R}^d$ *and let* $\mathcal{P}_\mathcal{C}$ *be the projection operator on* $\mathcal{C}$, *then for* $u, v \in \mathbb{R}^d$ *and any,* $c \in \mathcal{C}$ *we have*

$$||\mathcal{P}_\mathcal{C}(v + c) - \mathcal{P}_\mathcal{C}(u + c)|| \leq ||u - v|| \tag{44}$$

Given this, we have the following proof of Proposition.3

*Proof.* Consider the update

$$\begin{aligned}
\mathbf{w}^{t+1} &= \mathcal{P}_{\mathcal{C}_{\mathrm{seq}}}\big(\mathbf{w}^t - \eta_1\boldsymbol{g}_1^t\big), \quad \boldsymbol{g}_1^t = \mathcal{L}(\theta^*(\boldsymbol{w}, \boldsymbol{v}))\nabla_{\mathbf{w}} \ln p(\boldsymbol{w}|\mathbf{w}^t) \\
\mathbf{v}^{t+1} &= \mathcal{P}_{\mathcal{C}_{\mathrm{tkn}}}\big(\mathbf{v}^t - \eta_2\boldsymbol{g}_2^t\big), \quad \boldsymbol{g}_2^t = \mathcal{L}(\theta^*(\boldsymbol{w}, \boldsymbol{v}))\nabla_{\mathbf{v}} \ln p(\boldsymbol{v}|\boldsymbol{w} \odot \mathbf{v}^t).
\end{aligned} \tag{45}$$

Since the stochastic gradient mappings $\hat{\mathcal{G}}_{\mathbf{w}}^t$, $\hat{\mathcal{G}}_{\mathbf{v}}^t$ and deterministic gradient mappings denote as

$$\begin{aligned}
\hat{\mathcal{G}}_{\mathbf{w}}^t &= \frac{1}{\eta_1}(\mathbf{w}_t - \mathbf{w}_{t+1}), \quad \hat{\mathcal{G}}_{\mathbf{v}}^t = \frac{1}{\eta_1}(\mathbf{v}_t - \mathbf{v}_{t+1}); \\
\mathcal{G}_{\mathbf{w}}^t &= \frac{1}{\eta_1}\big(\mathbf{w}_t - \mathcal{P}_{\mathcal{C}_{\mathrm{seq}}}\big(\mathbf{w}^t - \eta_1\nabla_{\mathbf{w}}\Phi(\mathbf{w}^t, \mathbf{v}^t)\big)\big), \quad \mathcal{G}_{\mathbf{v}}^t = \frac{1}{\eta_2}\big(\mathbf{v}_t - \mathcal{P}_{\mathcal{C}_{\mathrm{tkn}}}\big(\mathbf{v}^t - \eta_2\nabla_{\mathbf{v}}\Phi(\mathbf{w}^t, \mathbf{v}^t)\big)\big).
\end{aligned} \tag{46}$$

Therefore

$$
\begin{aligned}
\Phi(\mathbf{w}^{t+1}, \mathbf{v}^{t+1}) \leq{} & \Phi(\mathbf{w}^t, \mathbf{v}^t) + \langle \nabla_{\mathbf{w}}\Phi(\mathbf{w}^t, \mathbf{v}^t), \mathbf{w}_{t+1} - \mathbf{w}_t \rangle \\
& + \langle \nabla_{\mathbf{v}}\Phi(\mathbf{w}^t, \mathbf{v}^t), \mathbf{v}_{t+1} - \mathbf{v}_t \rangle + \frac{L}{2}||\mathbf{w}_{t+1} - \mathbf{w}_t||^2 + \frac{L}{2}||\mathbf{v}_{t+1} - \mathbf{v}_t||^2 \\
\leq{} & \Phi(\mathbf{w}^t, \mathbf{v}^t) - \eta_1||\hat{\mathcal{G}}_{\mathbf{w}}^t||^2 - \eta_2||\hat{\mathcal{G}}_{\mathbf{v}}^t||^2 + \frac{L\eta_1^2}{2}||\hat{\mathcal{G}}_{\mathbf{w}}^t||^2 + \frac{L\eta_2^2}{2}||\hat{\mathcal{G}}_{\mathbf{v}}^t||^2 \\
& + \eta_1\langle \delta_1^t, \hat{\mathcal{G}}_{\mathbf{w}}^t \rangle + \eta_2\langle \delta_2^t, \hat{\mathcal{G}}_{\mathbf{v}}^t \rangle \ \ (\text{Lemma8}), \\
\leq{} & \Phi(\mathbf{w}^t, \mathbf{v}^t) - (\eta_1 - \frac{L\eta_1^2}{2})||\hat{\mathcal{G}}_{\mathbf{w}}^t||^2 - (\eta_2 - \frac{L\eta_2^2}{2})||\hat{\mathcal{G}}_{\mathbf{v}}^t||^2 + \eta_1\langle \delta_1^t, \mathcal{G}_{\mathbf{w}}^t \rangle + \eta_2\langle \delta_2^t, \mathcal{G}_{\mathbf{v}}^t \rangle \\
& + \eta_1||\delta_1^2||||\mathcal{G}_{\mathbf{w}}^t - \hat{\mathcal{G}}_{\mathbf{w}}^t|| + \eta_2||\delta_2^2||||\mathcal{G}_{\mathbf{v}}^t - \hat{\mathcal{G}}_{\mathbf{v}}^t|| \\
\leq{} & \Phi(\mathbf{w}^t, \mathbf{v}^t) - (\eta_1 - \frac{L\eta_1^2}{2})||\hat{\mathcal{G}}_{\mathbf{w}}^t||^2 - (\eta_2 - \frac{L\eta_2^2}{2})||\hat{\mathcal{G}}_{\mathbf{v}}^t||^2 \\
& + \eta_1\langle \delta_1^t, \mathcal{G}_{\mathbf{w}}^t \rangle + \eta_2\langle \delta_2^t, \mathcal{G}_{\mathbf{v}}^t \rangle + \eta_1||\delta_1^t|| + \eta_2||\delta_2^t|| \ \ (\text{Lemma9})
\end{aligned}
\tag{47}
$$

So we obtain

$$
\begin{aligned}
(\eta_1 - \frac{L\eta_1^2}{2})||\hat{\mathcal{G}}_{\mathbf{w}}^t||^2 + (\eta_2 - \frac{L\eta_2^2}{2})||\hat{\mathcal{G}}_{\mathbf{v}}^t||^2 \leq{} & \Phi(\mathbf{w}^t, \mathbf{v}^t) - \Phi(\mathbf{w}^{t+1}, \mathbf{v}^{t+1}) \\
\leq \eta_1\langle \delta_1^t, \mathcal{G}_{\mathbf{w}}^t \rangle + \eta_2\langle \delta_2^t, \mathcal{G}_{\mathbf{v}}^t \rangle + \eta_1||\delta_1^2|| + \eta_2||\delta_2^2||, & \\
\sum_{t=1}^{T}(\eta_1 - \frac{L\eta_1^2}{2})||\hat{\mathcal{G}}_{\mathbf{w}}^t||^2 + (\eta_2 - \frac{L\eta_2^2}{2})||\hat{\mathcal{G}}_{\mathbf{v}}^t||^2 \leq{} & \Phi(\mathbf{w}^1, \mathbf{v}^1) - \Phi(\mathbf{w}^{T+1}, \mathbf{v}^{T+1}) \\
\leq \sum_{t=1}^{T}\eta_1\langle \delta_1^t, \mathcal{G}_{\mathbf{w}}^t \rangle + \eta_2\langle \delta_2^t, \mathcal{G}_{\mathbf{v}}^t \rangle + \eta_1||\delta_1^t|| + \eta_2||\delta_2^t||. &
\end{aligned}
\tag{48}
$$

Since $\mathbb{E}\langle \delta_2^t, \mathcal{G}_{\mathbf{v}}^t \rangle = 0$ and $\mathbb{E}\langle \delta_2^t, \mathcal{G}_{\mathbf{w}}^t \rangle = 0$. Beyond this, we also have $\mathbb{E}||\delta_1^t|| = \mathbb{E}||\mathbf{g}_1^t - \nabla_{\mathbf{w}}\Phi(\mathbf{w}^t, \mathbf{v}^t)|| \leq \sigma^2$ and $\mathbb{E}||\delta_2^t|| = \mathbb{E}||\mathbf{g}_2^t - \nabla_{\mathbf{v}}\Phi(\mathbf{w}^t, \mathbf{v}^t)|| \leq \sigma^2$. Combine them and we obtain

$$
\frac{1}{T}\sum_{t=1}^{T}\mathbb{E}\Big(||\hat{\mathcal{G}}_{\mathbf{w}}^t||^2 + ||\hat{\mathcal{G}}_{\mathbf{v}}^t||^2\Big) \leq \frac{\Phi(\mathbf{w}^1, \mathbf{v}^1) - \Phi^*}{\min_{\eta \in \{\eta_1, \eta_2\}}(1 - L\eta/2)T} + \frac{\sigma^2}{\min_{\eta \in \{\eta_1, \eta_2\}}(1 - L\eta/2)}
\tag{49}
$$

Then finally, we can bound the expectation of the deterministic gradient norms:

$$
\begin{aligned}
\mathbb{E}||\mathcal{G}_{\mathbf{w}}^t||^2 \leq 2\mathbb{E}||\hat{\mathcal{G}}_{\mathbf{w}}^t||^2 + 2\mathbb{E}||\mathbf{g}_{\mathbf{w}}^t - \nabla_{\mathbf{w}}\Phi(\mathbf{w}^t, \mathbf{v}^t)|| \leq 2\mathbb{E}||\hat{\mathcal{G}}_{\mathbf{w}}^t||^2 + 2\sigma^2; \\
\mathbb{E}||\mathcal{G}_{\mathbf{v}}^t||^2 \leq 2\mathbb{E}||\hat{\mathcal{G}}_{\mathbf{v}}^t||^2 + 2\mathbb{E}||\mathbf{g}_{\mathbf{v}}^t - \nabla_{\mathbf{v}}\Phi(\mathbf{w}^t, \mathbf{v}^t)|| \leq 2\mathbb{E}||\hat{\mathcal{G}}_{\mathbf{v}}^t||^2 + 2\sigma^2
\end{aligned}
\tag{50}
$$

So

$$
\begin{aligned}
& \frac{1}{T}\sum_{t=1}^{T}\mathbb{E}\Big(||\mathcal{G}_{\mathbf{w}}^t||^2 + ||\mathcal{G}_{\mathbf{w}}^t||^2\Big) \\
\leq{} & \frac{2}{T}\sum_{t=1}^{T}\mathbb{E}\Big(||\hat{\mathcal{G}}_{\mathbf{w}}^t||^2 + ||\hat{\mathcal{G}}_{\mathbf{w}}^t||^2 + 2\sigma^2\Big) \\
\leq{} & 2\Big(\frac{\Phi(\mathbf{w}^1, \mathbf{v}^1) - \Phi^*}{\min_{\eta \in \{\eta_1, \eta_2\}}(1 - L\eta/2)T} + \frac{\sigma^2}{\min_{\eta \in \{\eta_1, \eta_2\}}(1 - L\eta/2)} + 2\sigma^2\Big) \\
\leq{} & 2\sigma^2\Big(2 + \frac{1}{\min_{\eta \in \{\eta_1, \eta_2\}}(1 - L\eta/2)}\Big)
\end{aligned}
\tag{51}
$$

$\square$

## Appendix.C: Implementation

The pipelines of our QCS and its transfer learning variant for targeted IT, are illustrated in Algorithm.1 and 2 . The code would be released.

---

**Algorithm 1** Quadratic Coreset Selection (QCS)

---

**Input**: A pre-trained LLM with parameter $\theta$, IT datasets $\mathcal{D}$.
**Parameter**: $\eta_1$, $\eta_2$, $\epsilon$, $\alpha$, $K$
**Output**: QCS solver with $\mathbf{w}^*$,$\mathbf{v}^*$, LLM with parameter $\theta^*$.

1: Initialize QCS by $w_i = \frac{K}{|\mathcal{D}|}, \forall i \in [K]$; $v_{i,j} = \alpha, \forall j \in [K_i]$.
2: Initialize LoRA parameter $\theta'$ with respect to $\theta$.
3: **while w,v** not converge **do**
4:     Draw a stochastic batch $\mathcal{D}_s \subset \mathcal{D}$.
5:     Sample sequence-level and token-level coresets of $\mathcal{D}_s$ via $\boldsymbol{w} \sim p(\boldsymbol{w}|\mathbf{w})$, $\boldsymbol{v} \sim p(\boldsymbol{v}|\boldsymbol{w} \odot \mathbf{v})$.
6:     $\forall i \in [K], j \in [K_i], v_{i,j} \leftarrow v_{i,j} + (-1)^{v_{i,j}}\epsilon$.
7:     Perform LoRA tuning to update $\theta'$ via $\boldsymbol{w}$,$\boldsymbol{v}$.
8:     Update $\mathbf{w}$, $\mathbf{v}$ via (33,34) with $\mathcal{D}_s$.
9: **end while**
10: Obtain $\mathbf{w}^* \leftarrow \mathbf{w}$, $\mathbf{v}^* \leftarrow \mathbf{v}$
11: Sample $K$ sequences by $\mathbf{w}^*$ as $\boldsymbol{w}^*$ to construct stochastic training batches;
12: **while** each training batch not converge **do**
13:     Sample tokens by $\mathbf{v}^*$ with respect to the sequences in the training batch, update $v_{i,j} \leftarrow v_{i,j} + (-1)^{v_{i,j}}\epsilon$;
14:     Update $\theta$
15: **end while**
16: **return** $\mathbf{w}^*$,$\mathbf{v}^*$, $\theta^* \leftarrow \theta$.

---

---

**Algorithm 2** QCS transfer learning variant

---

**Input**: A pre-trained LLM with parameter $\theta$, IT source datasets $\mathcal{D}_{\mathsf{train}}$, IT target datasets $\mathcal{D}_{\mathsf{val}}$.
**Parameter**: $\eta_1$, $\eta_2$, $\epsilon$, $\alpha$, $K$
**Output**: QCS solver with $\mathbf{w}^*$,$\mathbf{v}^*$, LLM with parameter $\theta^*$.

    Use Algorithm.1 to obtain $\mathbf{w}^*$,$\mathbf{v}^*$ by $\mathcal{D}_{\mathsf{train}}$;
    Initialize two separate LoRA module $\phi$,$\psi$;
    For $\mathcal{D}_{\mathsf{train}}$, generate the stochastic label $\boldsymbol{w}_i^*$ for each sequence, generate the stochastic label $\boldsymbol{v}_{i,j}^*$ for each token;
    For $\mathcal{D}_{\mathsf{train}}$, update $\phi$ to predict $\boldsymbol{w}_i^*$ for each sentence, update $\psi$ to predict $\boldsymbol{v}_{i,j}^*$ for each token;
    For $\mathcal{D}_{\mathsf{val}}$, apply test-time training to update update $\phi$ and update $\psi$;
    Sample $K$ sequences by $\phi$ as $\boldsymbol{w}^*$ to construct stochastic training batches;
    **while** each training batch not converge **do**
        Sample tokens by $\psi$ with respect to the sequences in the training batch, update $v_{i,j} \leftarrow v_{i,j} + (-1)^{v_{i,j}}\epsilon$;
        Update $\theta$
    **end while**
    **return** $\theta^* \leftarrow \theta$.

---

The experimental setup of regular IT refers to [38]. The experimental setup of targeted IT is followed.

**Training and evaluation setups** In the first experiment, we follow the training and evaluation settings in [60] and for the second experiment, we adopt the continual IT setup derived from [56] the task orders (1): QA $\rightarrow$ QG $\rightarrow$ SA $\rightarrow$ Sum. $\rightarrow$ Trans.; (2): Trans. $\rightarrow$ SA $\rightarrow$ QA $\rightarrow$ Sum. $\rightarrow$ QG; (3): Sum. $\rightarrow$ QG $\rightarrow$ Trans. $\rightarrow$ QA $\rightarrow$ SA. The continual learning order is as follows: (Warmup: Alpaca52k $\rightarrow$ task order. The experiment is conducted with 1% / 0.1% of the training data from SuperNI tasks and Alpaca-52k as demonstrations, respectively.

| Dataset | # Shot | # Tasks | $|\mathcal{D}_{\mathsf{val}}|$ | $|\mathcal{D}_{\mathsf{test}}|$ | Answer Type |
|---|---|---|---|---|---|
| **MMLU** | 5 | 57 | 285 | 18,721 | Letter options |
| **TYDIQA** | 1 | 9 | 9 | 1,713 | Span |
| **BBH** | 3 | 23 | 69 | 920 | COT and answer |
| **GSM-8K** | 8 | - | 10 | 1319 | COT and answer |

Table 4: The statistic of evaluation set.

**Training.** We implemented our QCS optimization strategy with LoRA incorporated a learning rate scheduler featuring linear warm-up and cosine decay, with the learning rate peaking at $2 \times 10^{-5}$. We standardized the training process across all selected datasets, utilizing a batch size of 128 and conducting 4 epochs to obtain the QCS parameters and the optimal coreset distribution parameters **v**, **w**. In the adaptation phase, we fix the LoRA parameters then fine-tune **v**, **w** with one epoch. The setup has been applied in targeted IT and continual IT experiments.

**Training setup of targeted IT.** The sizes revealed that extending the epoch count for smaller datasets did not yield significant performance enhancements. Consequently, we adopted a uniform epoch count across all experimental conditions. The LoRA module was configured with a rank of 128, an $\alpha$ parameter of 512, and a dropout rate of 0.1. We applied LoRA matrices to all attention matrices within the model architecture. This configuration resulted in trainable parameter counts of 135 million (1.95% of total parameters) for LLAMA-2-7B and 209 million (1.59%) for LLAMA-2-13B. To ensure robustness, each experimental condition was replicated thrice using distinct random seeds. For random selection methodologies, this entailed generating three unique subsets from the training corpus.

**Training setup of regular IT.** In **TruthfulQA**,**TruthfulQA**, **MMLU**, following [38], we use the instruction prompts in Vicuna [14] construction, then obtain the self-generated responses from the pre-trained **LLAMA-2** family models. We evaluate the base models trained with the instruction-response pairs through the three common instruction-following benchmarks. In AlpaEval experiment, LLM is trained with data selected by the mixture of FLAN V2, COT, DOLLY, and OPEN ASSISTANT 1 (More refers to the setup in targeted IT).

## Appendix.D: Complementary Experiments

### QCS *v.s.* LESS: more analysis on computation and memory cost

With regards to Table.1, the training time for LoRA in Algo.1 is almost consistent with the warmup stage in LESS, yet we need a labeling process (IO process) to prepare Algo.2 (it leads to over a half of computation cost in the warm-up stage in our implementation) and another LoRA in Algo.2. Note that the labeling procedure can be replaced by a teacher network that dynamically offers the soft label in Algo.2. This memory-time trade-off may further cuts off a significant amout of computation in GPU hours.

As for the convergence behavior, The charts demonstrate a clear and consistent convergence pattern for Algo2, where the validation loss in BBH exhibits a distinct U-shaped curve, reaching its minimum around epoch 1-2 before starting to increase due to overfitting. This convergence behavior reveals a crucial practical efficiency gain: optimal validation performance is achieved very early in training. This allows for early stopping, saving significant training time and computational resources beyond the per-step metrics in Table 2, thereby confirming the methods' practical efficiency.

### Efficient instruction-tuning for mathematical reasoning

Note that QCS is particularly impressive when the response is long in the selection data. It is supposed to be superior in mathematical reasoning tasks, so we provide a more challenging evaluation that focus on this problem.

**Setup.** We adopt a large reasoning model (LRM) as the backbone, specifically `Qwen-2.5-Math-7B-Instruct` [63]. Fine-tuning is performed with 10% long CoT reasoning responses selected from `OpenR1-Math-220k` [21] using our QCS sampler ($\epsilon = 0.1$, $\alpha = 0.2$). For comparisons under the regular instruction-tuning (IT) setup, we include: (i) the original `Qwen-2.5-Math-7B` (Base LRM), (ii) the model fine-tuned on 100% of `OpenR1-Math-220k` (Full), (iii) the model fine-tuned with 10% data randomly sampled from `OpenR1-Math-220k` (Random), and (iv) the model fine-tuned with 10% IDS-selected data (IDS).

**Training Details.** We use the system prompt "Please reason step by step, and put your final answer within \boxed{}." Decoding is greedy with temperature set to 0 and `top_p=1`, producing a single output. Evaluation follows the rule-based criteria of [64] , checking both numbers and formulas. We report Pass@1 (P@1), following [63]. All runs are conducted on a server with $4\times$A100-SXM4-80GB GPUs. We set the sequence length to 16,384 via RoPE scaling, batch size 1, gradient accumulation

4, and learning rate $5\mathrm{e}{-5}$ with warmup ratio 0.1 and cosine decay. Training lasts 3 epochs. For the judge model, we apply LoRA with rank 16, $\alpha = 32$, and dropout 0.1 for 1 epoch. After training, we evaluate on MATH [27] and GaoKao-MATH [63].

Table 5: Pass@1 (P@1) on MATH and GaoKao-MATH. Backbone: `Qwen-2.5-Math-7B-Instruct`.

| Benchmarks / Baselines | Base LRM | Full | Random | IDS | QCS |
|---|---|---|---|---|---|
| MATH | 0.842 | 0.894 | 0.878 | 0.889 | **0.901** |
| GaoKao-MATH | 0.781 | 0.783 | 0.789 | 0.863 | **0.878** |

**Results.** Table 5 reports P@1. Using only 10% of `OpenR1-Math-220k`, QCS surpasses Full on MATH (0.901 vs. 0.894) and shows clear improvements over Random and IDS. On GaoKao-MATH, QCS achieves 0.878, outperforming Full (0.783), Random (0.789), and IDS (0.863), demonstrating strong sample efficiency for complex, long-response reasoning.

### Case Analysis of Selected Instances

We illustrate some instances selected by our algorithm along with the tokens that were identified as "critical" during post-training.

### Case Study 1: Mathematical Reasoning (GSM8K)

**Instance A: A High-Value, Complex Problem (Selected and Prioritized by QCS)**

| | |
|---|---|
| **Problem:** | "A garden has 10 yellow flowers. There are 80% more purple flowers than yellow ones, and the number of green flowers is 25% of the yellow and purple flowers combined. How many flowers are in the garden?" |
| **Sequence-Level Analysis:** | QCS flags this as high-value because it requires multiple dependent steps and understanding percentages, a complex reasoning skill. |
| **Token-Level Analysis (QCS Advantage):** | Within the solution, QCS identifies and up-weights critical tokens that form the logical backbone of the reasoning process. |
| **Solution with Critical Tokens Emphasized:** | "First, calculate 80% of the **10** yellow flowers: $\mathbf{0.80 \times 10 = 8}$. The number of purple flowers is $\mathbf{10 + 8 = 18}$. Next, the total of yellow and purple is $\mathbf{10 + 18 = 28}$. Then, calculate 25% of that total: $\mathbf{0.25 \times 28 = 7}$. Finally, the total number of flowers is $\mathbf{10 + 18 + 7 = 35}$." |
| **Comparison:** | A sequence-level method would treat all tokens equally. QCS focuses training on crucial numbers (**10**, **18**, **28**, **35**), operators ($\times$, **+**), and percentages (**0.80**, **0.25**), helping the model learn the mathematical procedure, not just the narrative. |

### Case Study 2: Mathematical Reasoning (GSM8K)

**Instance B: A Simpler, Less Influential Problem (Down-weighted by QCS)**

| | |
|---|---|
| **Problem:** | "A bakery sold 15 cakes on Monday. On Tuesday, it sold 12 more cakes than on Monday. How many cakes were sold in total?" |
| **Sequence-Level Analysis:** | A standard method might select this for its basic arithmetic. QCS ranks it lower because the reasoning path is simpler and less informative for an already capable LLM. |
| **Token-Level Analysis:** | The critical tokens are basic (**15**, **+**, **12**, **=**, **27**). While correct, they do not introduce the same level of complexity as Instance A; QCS recognizes lower informational density. |

**Case Study 3: Complex Instruction Following (AlpacaEval / OpenAssistant)**

---

**Instance C: A High-Value, Complex Problem (Selected and Prioritized by QCS)**

| | |
|---|---|
| **Instruction:** | "Explain the main differences between nuclear fission and nuclear fusion, and list one practical application for each." |
| **Sequence-Level Analysis:** | QCS identifies this as a high-quality instruction because it requires factual accuracy, comparison, and structured knowledge. |
| **Token-Level Analysis (QCS Advantage):** | QCS prioritizes key concepts, relationships, and entities that must appear in a correct response. |
| **Solution with Critical Tokens Emphasized:** | "Nuclear **fission** is the **splitting** of a **heavy**, unstable nucleus into **two lighter nuclei**. A practical application is in **nuclear power plants**. In contrast, nuclear **fusion** is the process where two **light** atomic nuclei **combine** to form a heavier nucleus, releasing energy. A practical application is the energy source of **stars** like our Sun." |
| **Comparison:** | A sequence-level method may not differentiate between key terms and connecting language. QCS focuses the model on core vocabulary (**fission**, **fusion**, **splitting**, **combine**, **heavy**, **light**) and crucial entities (**nuclear power plants**, **stars**), ensuring factual and relational knowledge is learned correctly.. |

## Limitation and Future Work

QCS is sensitive to hyperparameters and stochasticity. Its performance hinges on budgets, token ratio, elastic weight, and policy learning rates; coupled with Bernoulli sampling, this can introduce high variance and unstable convergence—especially on smaller or less-stable backbones. The token-level selection offers limited practical memory or speed gains because pruned tokens often remain in context for next-token prediction; without architectural support (e.g., chunked attention or selective forwarding), compute savings are modest. Also, theoretical guarantees depend on smoothness, boundedness, and large-sample assumptions that can be violated in real LLM training (optimizer momentum, mixed precision), reducing the tightness of the bounds.

Beyond these, the transfer selectors (sequence/token heads) can be brittle under domain shift, miscalibrating selection probabilities across styles, lengths, or reasoning types; additional adaptation is often necessary. Engineering overhead remains nontrivial: although QCS avoids Hessian inverses, the hierarchical optimization, projections, selector training, and repeated sampling introduce IO and scheduling costs at scale. Together, these constraints highlight the need for automated hyperparameter tuning, variance reduction, architectural co-design for token pruning, stronger domain-adaptive selectors, and streamlined pipelines.

Overall, QCS pave the principled way to understand and investigate token-level information while selecting long-response instance for instruction-following. However, the hyper-parameter tuning and the complex training pipeline make it difficult to extend and further improvement. Some simple-yet-effective approaches inspired from Eq.5 can be more promising and expected.

