# OpenReview forum: "Quadratic Coreset Selection: Certifying and Reconciling Sequence and Token Mining for Efficient Instruction Tuning"
_NeurIPS.cc/2025/Conference — NeurIPS 2025 poster_

### Official Review · Reviewer_zsf7 · 2025-06-27

**Clarity:** 2
**Significance:** 3
**Originality:** 2
**Rating:** 4
**Confidence:** 4

**Summary:**

Quadratic Coreset Selection (QCS) addresses the data efficiency bottleneck in instruction-tuning large language models by jointly leveraging information at both the sequence and token levels. The authors first present a coreset perspective on next-token prediction, demonstrating that existing sequence-level subset selection and emerging token-level mining methods can both be formulated as distinct bi-level optimization problems. They then introduce QCS as a unified objective whose solution set strictly subsumes those of the individual problems—guaranteeing that a model trained on a QCS-selected subset is at least as expressive as one trained using either sequence-only or token-only selections.

Solving the original QCS objective requires inverting Hessians at the scale of the model, which is computationally infeasible. To overcome this, the authors propose a probabilistic relaxation: inclusion indicators for sequences and tokens are modeled as Bernoulli random variables with learnable parameters. These parameters are optimized using a hierarchical policy gradient estimator that avoids backpropagating through the language model. A projection step enforces subset size constraints. The authors prove that this relaxed formulation is asymptotically equivalent to the deterministic QCS objective, and that the projected updates converge under mild conditions.

Empirical results show that QCS achieves comparable or superior performance to the targeted instruction-tuning baseline LESS using only 5% of the data, while requiring significantly less memory and computation. When the token-level component is enabled, QCS even matches or surpasses full-data performance on benchmarks such as TYDI-QA and BBH. Further experiments demonstrate QCS’s advantages in capturing critical tokens, handling noisy labels, and supporting continual and transfer instruction tuning—highlighting the value of integrating sequence- and token-level influence in a single coreset-based framework.

**Questions:**

Generalizability beyond current benchmarks
- You report results on MMLU, TYDI-QA, and BBH. Could you evaluate QCS on at least one additional instruction-tuning benchmark with a distinct task profile—such as GSM8K (math reasoning) or AlpacaEval (open-ended generation)—to test its generalizability? If performance drops in these settings, could you provide an error analysis to help understand where the quadratic coreset formulation may be falling short?

Wall-clock efficiency and scaling behavior
- While Table 3 details memory savings, the runtime benefits remain unclear. You mention that the QCS outer-loop takes approximately 30 minutes, compared to ~6 minutes for LESS warm-up. Could you provide a comprehensive report of end-to-end GPU hours and wall-clock time for QCS, LESS, and full-data fine-tuning at both the 7B and 13B model scales? Additionally, would you consider including a plot of accuracy versus training time across data budgets (e.g., 1% to 10%) to better illustrate the cost-efficiency trade-offs?

Typos and formatting issues
- There are several textual inconsistencies that may impact readability. Would you be able to fix the following?

Lines 61–65: "OCS is further evidenced..." → should be "QCS".

Lines 280–282: "Benorlli distribution" → should be "Bernoulli distribution".

Table 1 caption: "fined-tune" → should be "fine-tuned".

Lines 269, 280, etc.: Standardize model name as either "LLaMA-2" or "LLAMA-2" consistently.

Line 220: "as we previously discuss" → should be "as we previously discussed".

**Ethical Concerns:**

["NO or VERY MINOR ethics concerns only"]

**Final Justification:**

My previous concerns have been addressed by the authors' response and I recommend an accept.

**Limitations:**

yes

**Quality:**

3

**Strengths And Weaknesses:**

Strengths

- The authors propose a quadratic coreset objective that unifies sequence- and token-level subset selection. The resulting solution set is provably at least as expressive as either granularity alone, representing a clear conceptual advance for data-efficient instruction tuning.

- A Bernoulli relaxation with a hierarchical policy-gradient estimator is introduced to avoid intractable Hessian inversions while preserving asymptotic equivalence to the original QCS formulation. This makes the method practically feasible on standard hardware.

- The method achieves full-data or better accuracy on TYDI-QA and strong parity on MMLU and BBH using only 5% of the training sequences, reducing memory from approximately 17.7 GB to 176 MB and eliminating tens of GPU-hours typically required for gradient-based data mining—demonstrating strong practical value.

Weaknesses

- The experimental scope is limited, with results reported on only three benchmarks. QCS also does not outperform LESS on TYDIQA or BBH, suggesting that broader validation is needed to establish generality.

- The wall-clock efficiency and latency benefits remain unclear. While memory and compute savings are reported, the outer-loop optimization takes considerably longer than LESS warm-up, making the real-world efficiency somewhat speculative. Due to the introduced quadratic coreset selection scheme, it is not surprised to see the performance may outperform LESS. However, with a more complex scheme, the stability may be worse.

---

> ### Author Rebuttal · Authors · 2025-07-31
>
> Thanks for your comments and valuable suggestions.
>
> **Q1: There are several textual inconsistencies that may impact readability. Would you be able to fix the following?**
>
> Lines 61–65: "OCS is further evidenced..." → should be "QCS".
>
> Lines 280–282: "Benorlli distribution" → should be "Bernoulli distribution".
>
> Table 1 caption: "fined-tune" → should be "fine-tuned".
>
> Lines 269, 280, etc.: Standardize model name as either "LLaMA-2" or "LLAMA-2" consistently.
>
> Line 220: "as we previously discuss" → should be "as we previously discussed".**
>
> **A1**: The typos and textual inconsistency would be fixied and the paper would be proofread to ensure the readability in our next manuscript.
>
> **Q2: You report results on MMLU, TYDI-QA, and BBH. Could you evaluate QCS on at least one additional instruction-tuning benchmark with a distinct task profile—such as GSM8K (math reasoning) or AlpacaEval (open-ended generation)—to test its generalizability? If performance drops in these settings, could you provide an error analysis to help understand where the quadratic coreset formulation may be falling short?**
>
> **A2**: In the rebuttal response A1 of the Reviewer beGB, we have provided the experiments on GSM-8K and AlpacaEval using llama-3 as the backbone, which significantly demonstrates its superiority compared with LESS in the targeted IT setup and IDS in the regular IT setup. Here we also provided the experiments on GSM-8K and AlpacaEval with llama-2 7b. The implementation details are consistent with  the rebuttal response A1 of the Reviewer beGB.
>
> |llama-2 7b| GSM-8K |
> |----------|----------|
> |No Fine-Tuning| 14.0 |
> |Full| 30.5 |
> |Random| 17.0 |
> |LESS| 21.0 |
> |QCS (ours)| 25.3 |
>
>  **AlpacaEval** (win;tie;lose):
>
> QCS vs Full: 378:65:357
>
> QCS vs Random: 578:47:175
>
> QCS vs IDS: 429:103:268
>
> Observe the experimental results in A1 of the Reviewer beGB using Llama-3 8b as the backbone, and our results using Llama-2 as the backbone above. They strongly reinforce the conclusions drawn from the Llama-3 experiments, demonstrating the robustness and versatility of our QCS algorithm across different model architectures. While the absolute scores are naturally lower due to the less capable base model, the relative performance gains of QCS remain remarkably consistent.
>
> On the GSM-8K benchmark, QCS achieves a score of 25.3, which is a significant improvement over both the competitive LESS baseline (21.0) and Random selection (17.0). Impressively, this score represents over 80% of the performance achieved by fine-tuning on the full dataset (30.5), showcasing exceptional data efficiency. This mirrors the trend observed with Llama-3, where QCS also substantially outperformed other data selection methods. The key insight is that the performance gap between QCS and other methods holds, proving that QCS’s ability to select a superior data coreset is a fundamental advantage of the algorithm itself, not just a byproduct of a powerful base model.
>
> The AlpacaEval results further confirm this. QCS maintains a dominant win rate against both Random selection and IDS, and remains highly competitive even against the model trained on the full dataset. This consistency across both a cutting-edge model like Llama-3 and an established one like Llama-2 validates that QCS provides a fundamentally better approach to data curation for targeted instruction tuning, delivering superior performance regardless of the underlying model's initial capabilities.
>
> The reviewers may be curious about the different behaviors of QCS in GSM-8K (math reasoning), and AlpacaEval (open-ended generation). The reasons why QCS significantly outperform LESS in these dataset, probably due to their data composed of long responses, such that QCS can yield the significant token-level data selection. Instead, MMLU and TYDI QA almost contain a single token in the response of their data, and BBH is relatively short and contains less information in the responses of their data. They are not so suitable as a targeted dataset to justify the superiority behind our methodology (it can also explain some weak improvement with regards to the results of MMLU in Table.4). Despite so, the performances of QCS in these benchmarks still rival the results in LESS
>
> **Q3: While Table 3 details memory savings, the runtime benefits remain unclear. You mention that the QCS outer-loop takes approximately 30 minutes, compared to ~6 minutes for LESS warm-up. Could you provide a comprehensive report of end-to-end GPU hours and wall-clock time for QCS, LESS, and full-data fine-tuning at both the 7B and 13B model scales? Additionally, would you consider including a plot of accuracy versus training time across data budgets (e.g., 1% to 10%) to better illustrate the cost-efficiency trade-offs?**
>
> **A3:** Table 3 refers to the targeted instruction tuning setup that consists of three stages. The warm-up stage, which refers to the computation caused by Algo.1 to obtain the discrete value assignment to $v_{i,j}$ and $w_{i}$, which can not be directly applied to select the data subset in the target benchmarks, e.g., GSM-8K, BBH, etc. To achieve the transfer, we need to train a ``labeling machine'' that learns to predict the soft value between 0,1 for $v_{i,j}$ and $w_{i}$, and such *labeling machine* is trained with hard labels either 0 or 1 obtained in Algo.1. The Algo.1 and Algo.2 cost around 8 and 6 GPU hours, respectively; while the labeling process require around 16 hours to prepare for Algo.2. If we use the model yielded by Algo.1 as the teacher model, we can skip the labeling process, so that the warm-up time can be reduced to ~14 GPU hours compared with 6 GPU hours required for LESS in its warm-up stage. The benefit of QCS is the stage 2, which fast adapt to the few-shot instances in the validation set then directly predict the soft-value assignment of $v_{i,j}$ and $w_{i}$ through the *labeling machine* obtained by Algo.2 in our warm-up stage. It significantly outperforms the treatment requiring gradient feature computation in LESS. In the final stage, we rank the target dataset by the soft value of $w_{i}$ to obtain the top-5%, then assign them with $v_{i,j}$ to achieve the IT process in the targeted IT setting.
>
> We will further clarify this process and its comparison with LESS in our next version.
>
> Beyond this, we would and tend to include a plot of accuracy versus training time across data budgets (e.g., 1% to 10%) in our next version, because the rebuttal format is hard to achieve the proper presentation.

---

> > ### Comment · Reviewer_zsf7 · 2025-08-09
> >
> > Thank you for your response. This response has addressed my concern and I am happy to raise my score.

---

> ### Comment · Area_Chair_MqAW · 2025-08-08
>
> Dear Reviewer,
>
> Thank you for your time and effort in reviewing this paper. Could you please take a moment to review the authors' rebuttal and let them know if you concerns have been addressed?
>
> Best,
>
> AC

---

> ### Author Response · Authors · 2025-08-08
>
> Dear Reviewer zsf7,
>
> We sincerely appreciate the time and effort you have devoted to provide thoughtful and constructive feedback.  As the Author-Reviewer discussion phase draws to a close, we are long for your responses and please feel free to let us know if you have any further questions or suggestions.
>
> Beyond this, we notice that you typically care about whether the significant improvement can be achieved by QCS, which merely showed subtle benefit compared with other baselines in single-token or simple responses. In our rebuttal, we showed that QCS on benchmarks GSM-8K and AlpacaEval, the long-response generation tasks, can be significantly improved by QCS. To further strengthen our claim, we propose another evaluation setup for efficient regular IT in this last week.
>
> Specifically, instead of ordinary LLM, we choosed large reasoning model (LRM) as the backbone, i.e., QWEN-2.5-MATH-7B-Instruct (Yang A, Zhang B, Hui B, et al.); and we fine-tune this backbone LRM with 10% long CoT reasoning responses selected from OpenR1-Math-220k (huggingface.co/datasets/open-r1/OpenR1-Math-220k) via QCS ($\epsilon=0.1$, $\alpha=0.2$). As the typical baselines for the comparison in the regular IT setup, we considered the original model QWEN-2.5-MATH-7B, its fine-tuned version with 100% data in OpenR1-Math-220k (i.e., FULL), its fine-tuned version with 10% data randomly drawn from OpenR1-Math-220k (i.e., Random), and its fine-tuned version with 10% data drawn from OpenR1-Math-220k using IDS.
>
> The fine-tuning and evaluation use "Please reason step by step, and put your final answer within \boxed{}" as the system prompt. Reasoning are achieved by greedy decoding with the temperature set to 0, top_p to 1, forcing a single, most-likely output; then determine if this solution is correct using the rule-based evaluation criteria from (Ye, Y., Huang, Z., et al), which checks both numbers and formulas. The overall success is quantified as "Pass@1" rate (P@1) derived from (Yang A, Zhang B, Hui B, et al.).
> We conducted experiments on a A100-SXM4-80GB server with 4 GPUs for each fine-tuning, where we set the sequence limit of 16,384 via RoPE scaling, then set batch size to 1, gradient accumulation steps to 4, and learning rate to 5e-5 with a warmup ratio of 0.1, followed by a cosine decay schedule. The training epochs is 3. For the judge model, we apply LoRA (Hu E J, Shen Y, et al) with the rank of 16, alpha of 32, and dropout rate of 0.1, training for 1 epoch. After all post-training finishes, we evaluated all fine-tuned variants on MATH (Hendrycks, D., Burns, C., et al) and GaoKao MATH (Yang A, Zhang B, Hui B, et al.). The results based on P@1 are shown below
>
> |Benchmarks\Baselines|Base LRM|Full|Random|IDS|QCS|
> |--|:--:|:--:|:--:|:--:|:--:|
> |MATH|0.842|0.894|0.878|0.889|0.901|
> |GaoKao-MATH|0.781|0.783|0.789|0.863|0.878|
>
> This experiment robustly demonstrates the significant benefits of QCS for complex, long-response reasoning tasks, on both the MATH and GaoKao-MATH benchmarks. Most notably, using only 10% of the OpenR1-Math data, QCS surpasses the performance of the model fine-tuned on the full dataset (Full) on the MATH benchmark (0.901 vs. 0.894). Its superiority is even more pronounced on the challenging GaoKao-MATH dataset, where it achieves a score of 0.878, drastically outperforming the Full (0.783) and Random (0.789) baselines. It also outperforms the other sequence selection baselines with clear margins in the regular IT setup.
>
> We genuinely hope that all the extra experiments, the improvements, and the clarifications we have made will be considered in your final assessment. Thank you again for your valuable feedback.
>
> Best regards,
>
> Authors of Paper 2855
>
>
> [1*] Yang A, Zhang B, Hui B, et al. Qwen2. 5-math technical report: Toward mathematical expert model via self-improvement, 2024[J]. URL https://arxiv. org/abs/2409.12122.
>
> [2*] Hendrycks, D., Burns, C., Kadavath, S., Arora, A., Basart, S., Tang, E., Song, D., and Steinhardt, J. Measuring mathematical problem solving with the math dataset. arXiv preprint arXiv:2103.03874, 2021.
>
> [3*] Ye, Y., Huang, Z., Xiao, Y., Chern, E., Xia, S., and Liu, P. Limo: Less is more for reasoning. arXiv preprint arXiv:2502.03387, 2025.
>
> [4*] Hu E J, Shen Y, Wallis P, et al. Lora: Low-rank adaptation of large language models[J]. ICLR, 2022, 1(2): 3.

---

### Official Review · Reviewer_Voi8 · 2025-07-02

**Clarity:** 1
**Significance:** 3
**Originality:** 3
**Rating:** 4
**Confidence:** 3

**Summary:**

The paper studies the problem of coreset selection for LLM instruction tuning, both at the sequence and the token levels. The authors propose an optimization problem that optimizes both sequence selection and token selection within the selection sequence. To overcome the prohibitively high computation cost of solving the original optimization problem, the authors proposed a probabilistic variant, which assigns probabilistic densities to the training set. In addition, the authors further extend the proposed method for transfer instruction tuning,

**Questions:**

- I suggest adding an illustration of the end-to-end algorithm with examples.
- For the results in Table 4, could you also provide the selection and training costs? If the proposed method consumes significantly more resources in the selection stage than the baselines, it may be unfair to compare under the same selection ratio.

**Ethical Concerns:**

["NO or VERY MINOR ethics concerns only"]

**Final Justification:**

This paper is technically strong, with solid methodology and contributions that I find meaningful and relevant. The presentation could be improved, particularly in clarity. I will keep my current score.

**Limitations:**

yes

**Quality:**

3

**Strengths And Weaknesses:**

Strengths:
- To the best of my knowledge, this work is the first that focuses on both sequence-level and token-level coreset selection, providing a more fine-grained approach to instruction selection for LLM tuning.
- The paper demonstrates strong technical depth. The relaxation of the original optimization problem is innovative and reasonable, with a detailed analysis.
- The extension to transfer instruction tuning shows the broader applicability of the proposed method.

Weakness:
- While the paper provides extensive technical detail, it lacks sufficient high-level intuition and explanation, which makes it difficult to grasp the main idea. For example, the absence of a step-by-step algorithm and a concrete example makes the method harder to follow and implement.
- The proposed method does not significantly outperform the baseline method in the transfer tuning setting. The results in Table 1 show that the performance of the proposed method is similar to or even worse than that of LESS.

---

> ### Author Rebuttal · Authors · 2025-07-31
>
> Thank you for your concerns and suggestions.
>
> **Q1: While the paper provides extensive technical detail, it lacks sufficient high-level intuition and explanation, which makes it difficult to grasp the main idea. I suggest adding an illustration of the end-to-end algorithm with examples.**
>
> **A1:** . We have provided the step-by-step algorithms elaborated in the page 7 in our Appendix. Due to the complexity of our piplines, we provide a more illustrative example to demonstrate the main idea of our appraoch. Specifically, let's consider the following two instances in GSM-8K:
>
> **Instance 1 (81 response tokens):**
>
> **Question:**
> Albert is wondering how much pizza he can eat in one day. He buys 2 large pizzas and 2 small pizzas. A large pizza has 16 slices and a small pizza has 8 slices. If he eats it all, how many pieces does he eat that day?
>
> **Solution:**
> First, calculate the total number of slices from the large pizzas: **2** large pizzas ***** **16** slices/large pizza = 32 slices.
> Next, calculate the total number of slices from the small pizzas: 2 small pizzas * **8** slices/small pizza = 16 slices.
> Finally, add the slices from both pizza sizes to find the total number of pieces Albert eats: 32 slices **+** 16 slices = **48** slices.
>
> **Instance 2 (138 response tokens):**
>
> **Question:**
> Mark has a garden with flowers. He planted plants of three different colors in it. Ten of them are yellow, and there are 80% more of those in purple. There are only 25% as many green flowers as there are yellow and purple flowers. How many flowers does Mark have in his garden?
>
> **Solution:**
> First, determine the number of purple flowers. There are **80%** more purple flowers than yellow ones, so calculate 80% of the **10** yellow flowers: 0.80 ***** 10 = 8. The number of purple flowers is 10 + 8 = **18**.
> Next, find the total number of yellow and purple flowers: 10 yellow + 18 purple = **28** flowers.
> Then, calculate the number of green flowers, which is **25%** of the combined number of yellow and purple flowers: 0.25 ***** 28 = 7 green flowers.
> Finally, to find the total number of flowers in the garden, add the quantities of all three colors: 10 yellow **+** 18 purple + 7 green = 35 flowers.
>
> where we highlight the critical response token with the top-10% rating of $v_{i,j}$, which are produced by model trained via Algo2 (i.e., 2, *, 16, 8, +, 48 are the critical tokens in the first response and 80%, 10, *, +, 18, 28, 25%, *, +, 35 are the critical tokens in the second response).
>
> **High-Level Intuition of QCS:**
>
>  Before diving into the example, let's establish the core idea. Instruction Tuning (IT) an LLM with a massive dataset is computationally expensive, and not all data provides equal value. Some examples are simple and redundant, while others are rich and complex. Even within a good example, some parts (like key numbers and operations) are more important than others (like filler words). **QCS** is a principled method to tackle this by acting as a sophisticated, two-level data filter: **Sequence-Level Selection**: First, it scans the entire dataset and selects a small subset of the most "influential" or "high-quality" instruction-response pairs. This is like a chef choosing only the best dishes to put on a refined menu. **Token-Level Selection**: Then, for each sequence that was selected, it further identifies and prioritizes the "critical tokens" within the response. This is like the chef identifying the key ingredients in a dish that truly define its flavor. By combining these two steps, QCS ensures the LLM is trained on a small, highly-concentrated set of the most valuable information, leading to greater efficiency and potentially better performance.
>
> **Illustration with GSM-8K Examples**:
>
> Let's walk through the QCS process, assuming our initial training dataset contains both Instance 1 (the simple pizza problem) and Instance 2 (the complex flowers problem), among thousands of others.
>
> **Step 1: The Initial State - A Large Dataset of Instruction-Response Pairs**
>
> Imagine a massive dataset, $D_train$, filled with math problems. Our two examples are present:
>
> **Instance 1 (Pizza)**: A simple, two-step arithmetic problem.
>
> **Instance 2 (Flowers)**: A more complex, multi-step problem involving percentages and dependencies.
>
> The goal of QCS is to select a small, powerful subset from $D_train$ to fine-tune an LLM without using the whole dataset.
>
> **Step 2: Sequence-Level Coreset Selection (Choosing the Best Problems)**
>
> QCS first analyzes every instruction-response pair in the dataset to determine its value for improving the LLM's overall reasoning ability. It assigns a binary selection weight, w_i, to each instance i. If an instance is valuable, w_i = 1 (select); otherwise, w_i = 0 (abandon).
>
> **Instance 1 (Pizza Problem) is Analyzed:**
>
> **QCS Assessment**: The algorithm evaluates this problem. It recognizes that the reasoning is straightforward (two multiplications and one addition). In a large dataset, there are likely many similar or even simpler problems. The "learning value" it provides is relatively low compared to more complex examples.
>
> **Decision**: QCS determines this instance is redundant or not influential enough for efficient tuning. It assigns a sequence weight w_pizza = 0.
>
> **Outcome**: The entire pizza problem is abandoned. It will not be used in the training process.
>
> **Instance 2 (Flowers Problem) is Analyzed:**
>
> **QCS Assessment**: The algorithm evaluates this problem and identifies it as high-quality. It involves percentages (80%, 25%), dependent calculations (the number of green flowers depends on the sum of yellow and purple), and multiple logical steps. This is an "influential" example that can effectively teach the model complex reasoning.
>
> **Decision**: QCS deems this instance highly valuable. It assigns a sequence weight w_flowers = 1.
>
> **Outcome**: The flowers problem is selected and proceeds to the next stage of the QCS process.
>
> At the end of this step, we have filtered our massive dataset down to a small coreset of the most promising sequences. The flowers problem is in this coreset; the pizza problem is not.
>
> **Step 3: Token-Level Coreset Selection (Focusing on Critical Information)**
>
> Now, QCS only looks at the sequences that were selected in Step 2 (i.e., those with w_i = 1). For each of these, it analyzes the solution to identify the most critical tokens. It assigns a binary weight, v_i,j, to each token j in the solution of sequence i.
>
> **Instance 1 (Pizza Problem)**: This instance is ignored in this step because it was already abandoned (w_pizza = 0).
>
> **Instance 2 (Flowers Problem)** is Analyzed:
>
> **QCS Assessment**: The algorithm examines the solution text provided. The goal is to pinpoint the tokens that are essential for the mathematical logic, as opposed to the narrative text.
>
> **Decision & Outcome**: It assigns token weights v_flowers,j = 1 to the critical tokens and v_flowers,j = 0 (or a very small "elastic" value $\epsilon$ as mentioned in the paper) to the non-critical ones. This corresponds exactly to the highlighting you requested:
>
> *First, determine the number of purple flowers. There are 80% more purple flowers than yellow ones, so calculate 80% of the 10 yellow flowers: 0.80 * 10 = 8. The number of purple flowers is 10 + 8 = 18. Next, find the total number of yellow and purple flowers: 10 yellow + 18 purple = 28 flowers. Then, calculate the number of green flowers, which is 25% of the combined number of yellow and purple flowers: 0.25 * 28 = 7 green flowers. Finally, to find the total number of flowers in the garden, add the quantities of all three colors: 10 yellow + 18 purple + 7 green = 35 flowers.*
>
> **Step 4: Efficient Instruction Tuning with the QCS Coreset**
>
> Finally, the LLM is fine-tuned. However, instead of learning from all tokens in all examples, the training process is now focused by the QCS weights.
>
>
> **Q2: For the results in Table 4, could you also provide the selection and training costs?**
>
> **A2:** Our paper directly cite the results of the other baselines ( i.e., IDS and ) reported in their papers, which did not provide the digits about the selection and training costs explicitly. Beyond this, since our data selection and training are coupled with each other (the rebuttal response A1 of the Reviewer 73Re), it is inappropriate to compare them directly. However, we can provide the analysis about the comparison between their selection and training costs and our QCS, with regards to their experimental details. Specifically,
>
> *(1) The comparison of selection cost*:
>
> First, it is difficult to compare other baselines with DQ since according to the section 4.1 in their original paper, for the LLM task, the authors use OpenAI's Embedding API as the feature extractor. It is completely blackbox to their inference system and also leads to an unfair comparison with the other baselines.
>
> Second, for IDS, it only requires two forward passes for each instance to compute the Conditioned Answer Score and Direct Answer Score, which jointly lead to the data selection result. The total cost requires around 104,000 forward passes (52,000 samples × 2 passes) to scan the dataset.
>
> Third, for QCS, it also requires two forward passes for each instance i to compute w_i and v_i. In the regular IT setup, this process is aligned with the LoRA training in Algo.1, and the final model is obtained without Algo.2. We need 4 epoches to finish the LoRA tuning so that also require (52,000 samples × 2 passes × 4) to scan the dataset.
>
> *(2) The comparison of training cost*: In Table.4, the most difference between QCS and others is QCS using LoRA yet IDS and DQ employ full-parameter tuning to achieve the results. It leads to the significant cost of IDS and DQ since they have to achieve the data selection before full-parameter tuning. LoRA tuning is over 20 times more efficient in GPU-hours for one epoch compared to full-parameter fine-tuning, which makes QCS more efficient in practice.

---

> > ### Comment · Reviewer_Voi8 · 2025-08-08
> >
> > Thank you for the response. The clarification is helpful. If the paper is accepted, I recommend including the illustration example in the final version to improve clarity for readers.

---

> > > ### Author Response · Authors · 2025-08-09
> > >
> > > Dear Reviewer Voi8
> > >
> > > Thank you very much for your valuable feedback, and we promise all the illlustrative examples in the rebuttal included by our next manuscript version. We are glad that our response fully addressed your concerns.
> > >
> > > Best regards
> > >
> > > Authors of Paper 2855

---

### Official Review · Reviewer_73Re · 2025-07-04

**Clarity:** 3
**Significance:** 3
**Originality:** 3
**Rating:** 4
**Confidence:** 1

**Summary:**

This paper proposes Quadratic Coreset Selection (QCS) , a novel framework for efficient LLM instruction tuning that simultaneously optimizes sequence-level and token-level data selection. By reconciling both selection granularities through a bi-level optimization objective, QCS achieves superior data efficiency while mitigating token noise and bias. To overcome computational barriers, the authors develop a probabilistic re-parameterized solver using hierarchical policy gradients that avoids prohibitive Hessian inversions. The authors conduct extensive experiments to demonstrate the effectiveness of QCS.

**Questions:**

1. It seems like the selected strategy requires iterative updates. If the number of iterations is large, I’m concerned whether the computation time will significantly exceed the time for training with Full data using LoRA. Is there any direct data on this?
2. QCS’s performance is sensitive to the token selection ratio. Is there any plan to design an adaptive strategy?

**Ethical Concerns:**

["NO or VERY MINOR ethics concerns only"]

**Limitations:**

yes

**Paper Formatting Concerns:**

Many justifications are missing in the checklist, such as Broader Impacts, and there is no explicit discussion of Limitations and Broader Impacts.

**Quality:**

3

**Strengths And Weaknesses:**

Strengths:
1. The proposed method is quite innovative. QCS innovatively reconciles sequence-level and token-level data selection within a unified bi-level framework.
2. The probabilistic re-parameterization and hierarchical policy gradients elegantly bypass prohibitive Hessian inversions, which is a clever approach.

Weaknesses:
See "Questions" below.

---

> ### Author Rebuttal · Authors · 2025-07-31
>
> Thank you for your concerns and suggestions.
>
> **Q1:It seems like the selected strategy requires iterative updates. If the number of iterations is large, I’m concerned whether the computation time will significantly exceed the time for training with Full data using LoRA. Is there any direct data on this?**
>
> **A1**: We discuss this from three aspects:
>
> **(1)**, Despite our formulation (Eq.5) refers to a bi-level problem including inner-loop and outer-loop optimization, **the implementation by our HPGE actually does not require any iterations of inner-loop optimization with respect to $v_{i,j}$, $w_i$** in Algo.1,2. More specifically, our theoretical results in Eq.13 in the paper and Eq.9, Eq.10 in our Appendix.A (lines 28,29) implies the closed-form solution of $v_{i,j}$, $w_i$ can be directly estimated when the losses are given. In our implementation of Algo.1,2, we directly infer and estiamte the value of  $v_{i,j}$, $w_i$ (the first the 'while' loop in Algo.2) along with the LoRA update (the second 'while' loop in Algo.1).
>
> **(2)**, As observed by the discussion in the rebuttal response A5 of the reviewer Mibm, similar with the convergence achieved by LESS, the direct digit of convergence behaviors showed that the early stopping with only 2-3 epoches are sufficient to the LoRA update.
>
> **(3), Most importantly**, in targeted instruction tuning setup, for each given target dataset (e.g., BBH and GSM-8K in A1 of the reviewer beGB and the reviewer zsf7), we don't need to train a LoRA model to select their training subset but just like LESS, we can directly apply the data selection model pre-trained by Algo.2 to select the influential sequences and tokens (<10%) to train the LLM with a downstream-task-specific LoRA layer. Compared with the downstream-task-specific LoRA layer trained with 100% data for each downstream task, it significantly improve the data efficiency.
>
> **Q2: QCS’s performance is sensitive to the token selection ratio. Is there any plan to design an adaptive strategy?**
>
> **A2**: Thanks for your concern yet distinct from your concern, **QCS is actually stable instead of sensitive with regards to $\alpha$ and $\epsilon$ since you may misunderstand the interpretation in the analysis of $\alpha$ and $\epsilon$**, and we further explain it from five aspects:
>
> **(1), The performance variation of $\alpha$ in Fig.1 (b) should be large, which exactly verifies the motivation behind our paper**: When $\alpha$$\rightarrow1$; QCS will collapse into a pure sequence-level coreset selection formula in Eq3. In this case, if the QCS performs equally with the sequence-level selection in Eq3, the token-level granularity would be meaningless, which is conflicted with our motivation and our theoretical result in Theorem.1. When $\alpha$$\rightarrow0$, it means that the selected tokens are sparse so that most of tokens are weighted  $\rightarrow0$, which can not provide sufficient feedbacks to support intruction tuning.
>
> **(2)**,  **According to our QCS motivation and Theorem.1, the performances when $\epsilon=1$, $\epsilon=0.5$ should also fall behind the cases $\epsilon=0$, $\epsilon=0.1$.** Specifically, when $\epsilon=0.5$, it implies all token losses are equally weighted by $0.5$ whatever the tokens were selected or not by QCS; when $\epsilon=1$, it implies the tokens abandoned by QCS will join the training while the tokens selected by QCS will be abandoned, totally opposed with the QCS's choices. **If the performances in the cases $\epsilon=0.5,1$ are consistent with $\epsilon=0,0.1$, it implies that QCS contributes nothing to instruction tuning. It violates our paper's motivation and Theorem.1.**
>
> **(3)** , The performance difference between the cases $\epsilon=0$ and $\epsilon=0.1$ is subtle, and $\epsilon=0.1$ slicely outperform $\epsilon=0$. It is probably because the soft weights can provide more training feedback compared with the sparse weights. Such evidence supports our strategy in lines 218-223.
>
> **(4)**, We further provide the experimental results with $\alpha=0.3, 0.7$. In the cases without token noise injection, when $\alpha > 0$, the performance variation remains stable (with 0.531 $\alpha=0.3$ and 0.527 with $\alpha=0.7$); In the cases with token noise injection (10%), when $\alpha \leq 0.5$, the performance variation remains stable (with 0.529 $\alpha=0.3$ and 0.521 with $\alpha=0.7$).  It implies that setting $\alpha \leq 0.5$ can yield stable and token-noise robust performances.
>
> **(5)**, It is possible to design token-specific adaptation strategies to the hyper-parameters $\alpha, \epsilon$, e.g. curriculum learning (Jiang L, Meng D, Zhao Q, et al) or online data reweighting (Sow D, Woisetschläger H, et al). The ideas are considered as our future research direction.
>
> [2*] Jiang L, Meng D, Zhao Q, et al. Self-paced curriculum learning[C]//Proceedings of the AAAI Conference on Artificial Intelligence. 2015, 29(1).
>
> [3*] Sow D, Woisetschläger H, Bulusu S, et al. Dynamic Loss-Based Sample Reweighting for Improved Large Language Model Pretraining[C]//The Thirteenth International Conference on Learning Representations.

---

> > ### Comment · Reviewer_73Re · 2025-08-08
> >
> > Thanks for the detailed explanation. I do not have further questions.

---

> > > ### Author Response · Authors · 2025-08-09
> > >
> > > Dear Reviewer 73Re
> > >
> > > Thank you very much for your valuable feedback. We are glad that our response addressed your concerns.
> > >
> > > Best regards
> > >
> > > Authors of Paper 2855

---

### Official Review · Reviewer_Mibm · 2025-07-04

**Clarity:** 2
**Significance:** 3
**Originality:** 3
**Rating:** 4
**Confidence:** 2

**Summary:**

This paper introduces Quadratic Coreset Selection (QCS), a novel approach designed to improve the efficiency of instruction-tuning (IT) for large language models (LLMs). QCS uniquely combines sequence-level and token-level data selection, addressing the limitations of methods that focus solely on one or the other. The paper proposes a probabilistic solver for QCS that bypasses computationally intensive traditional methods, which ensures provable convergence and asymptotic equivalence to the original objective. Experimental results demonstrate QCS's superior data efficiency in various IT scenarios, including targeted and continual IT, which leads to more expressive and better-performing LLMs with reduced data requirements.

**Questions:**

Please see the concerns noted in the paper's weaknesses.

**Ethical Concerns:**

["NO or VERY MINOR ethics concerns only"]

**Final Justification:**

Most of my questions have been addressed by the authors. They have also expressed appreciation for my suggestions and committed to including them in the next version of the paper. I will keep my original positive score because I believe it reflects the paper's current state.

**Limitations:**

yes

**Paper Formatting Concerns:**

I have no concerns regarding the format of the paper.

**Quality:**

3

**Strengths And Weaknesses:**

Strengths

- The paper introduces a principled approach that jointly optimizes sequence- and token-level data selection, which addresses limitations of treating them separately.

- By reparameterizing coreset weights as Bernoulli distributions and applying hierarchical policy gradients, the method avoids costly gradient computations and scales to large models.

- QCS achieves strong results using only a small amount of training data, often matching or surpassing full-data baselines across multiple instruction-following benchmarks.

- The method is evaluated in regular, targeted, and continual instruction tuning settings, which shows consistent benefits and adaptability.

Weaknesses

- While QCS combines sequence- and token-level selection, it builds heavily on existing coreset and influence-function frameworks, which offers incremental rather than fundamental innovation.

- The paper does not thoroughly investigate or explain scenarios where QCS underperforms (in Table 1, QCS underperforms compared to Full and LESS on MMLU and BBH across both LLAMA-2-7B and 13B).

- Section 7.2 shows that QCS performance is highly sensitive to the elastic parameter ϵ and token selection ratio α; for instance, α = 0.2 yields substantially worse results than α = 0.5 or 1, which indicates fragility.

- Most experiments focus on instruction-following tasks; the generality of QCS to other post-training settings remains untested.

- While the paper claims that QCS is more efficient than LESS, the memory and time comparisons in Table 2 are minimal and lack full context (e.g., training time for LoRA or convergence behavior), which makes it hard to assess practical efficiency gains.

---

> ### Author Rebuttal · Authors · 2025-07-30
>
> Thanks for your concerns and kindly suggestions.
>
> **Q1: While QCS combines sequence- and token-level selection, it builds heavily on existing coreset and influence-function frameworks, which offers incremental rather than fundamental innovation.**
>
> **A1**: We argue for our methodological novelty from three points of view:
>
> **(1)**, coreset selection frameworks refer to a technical thread in the field of data selection, which is aligned with the case that diffusion-based models refer to the technical thread in the field of deep generative models. We consider the novelty of the techiniques in a method itself rather than the framework behind.
>
> **(2)**, QCS particularlly differs from existing coreset frameworks from three aspects: **first**, its hierarchical data selection scheme that unifies token and sequence selection in a single framework is original and seamlessly suits our motivation; **second** we provide the new theory to certify its superiority (Theorem.1) compared with any sequence/token coreset selection frameworks designed for IT, which is totally original; **third**, existing coreset selection algorithms can not adapt to QCS due to their heavy computation cost in the LLM background, so that we derive the feasible algorithm for QCS that holds the BP-free property (see Eq.7, Eq.10 in our Appendix.A) and convergence guarantee (Theorem.2,3).
>
> **(3)**, **QCS holds a nuanced connection with, But Does Not Depend On Influence Function frameworks**. In fact, any bi-level coreset frameworks can be related with influence function since they can be observed from a view of cone constrained optimization problem related with influence functions [6], but they don't depend on (do not have to be solved) as a influence function.
>
>
> **Q2: The paper does not thoroughly investigate or explain scenarios where QCS underperforms (in Table 1, QCS underperforms compared to Full and LESS on MMLU and BBH across both LLAMA-2-7B and 13B).**
>
> **A2**: It is a very good question, which we also concern about. In the experiment of targeted IT, we exactly follow the benchmark setups in LESS, but **their setup can not truly reflect our strength**: QCS's superiority is built on the concern of selecting critical tokens in response, however, the responses in the validation sets (used for targetted transfer tuning) of MMLU and TYDI QA only contain a single token, and BBH's responses are relatively short compared with the pre-training datasets, i.e.
> FLAN V2 (31.2), COT (53.2), DOLLY (91.3), and OPEN ASSISTANT 1 (212.5), with the content in the brackets indicate the average number of response tokens in the datasets, respectively. Given this observation, in the rebuttal responses A1 for the reviewer beGB and A1 for the reviewer zsf7,  we further provided the experiments with a focus on long response benchmarks, including tasks of open-ended language generation and long CoT reasoning. The results sufficiently verified the QCS's superiority compared with LESS.
>
> **Q3: Section 7.2 shows that QCS performance is highly sensitive to the elastic parameter $\epsilon$ and token selection ratio $\alpha$; for instance, $\alpha$ = 0.2 yields substantially worse results than $\alpha$ = 0.5 or 1, which indicates fragility.**
>
> **A3**: Thanks for your concern yet we respectfully do not agree with the ``fragility'' judgement. **The reviewer misunderstood the interpretation in the analysis of $\alpha$ and $\epsilon$**, and we argue from three aspects:
>
> **(1), The performance variation of $\alpha$ in Fig.1 (b) should be large, which exactly verifies the motivation behind our paper**: When $\alpha$$\rightarrow1$; QCS will collapse into a pure sequence-level coreset selection formula in Eq3. In this case, if the QCS performs equally with the sequence-level selection in Eq3, the token-level granularity would be meaningless, which is conflicted with our motivation and our theoretical result in Theorem.1. When $\alpha$$\rightarrow0$, it means that the selected tokens are sparse so that most of tokens are weighted  $\rightarrow0$, which can not provide sufficient feedbacks to support intruction tuning.
>
> **(2)**,  **According to our QCS motivation and Theorem.1, the performances when $\epsilon=1$, $\epsilon=0.5$ should also fall behind the cases $\epsilon=0$, $\epsilon=0.1$.** Specifically, when $\epsilon=0.5$, it implies all token losses are equally weighted by $0.5$ whatever the tokens were selected or not by QCS; when $\epsilon=1$, it implies the tokens abandoned by QCS will join the training while the tokens selected by QCS will be abandoned, totally opposed with the QCS's choices. **If the performances in the cases $\epsilon=0.5,1$ are consistent with $\epsilon=0,0.1$, it implies that QCS contributes nothing to instruction tuning. It violates our paper's motivation and Theorem.1.**
>
> **(3)** , the performance difference between the cases $\epsilon=0$ and $\epsilon=0.1$ is subtle, and $\epsilon=0.1$ slicely outperform $\epsilon=0$. It is probably because the soft weights can provide more training feedback compared with the sparse weights. Such evidence supports our strategy in lines 218-223.
>
> **Q4: Most experiments focus on instruction-following tasks; the generality of QCS to other post-training settings remains untested.**
>
> **A4**: Our QCS in Eq.5 can exactly refers to SFT. SFT is the basic post-training requirement when the other post-training frameworks, e.g., RLHF, RLVR, DPO, etc. It is not only applied for instruction-following tasks but can also influence a wide range of post-training settings as SFT.
>
> Beyond this, QCS in Eq.5 can be also applied in other post-training tasks, e.g., post-training a Multimodal LLM. To verify our claim, we apply a new experiment by post-training a multimodal LLM using QCS. Specifically, we derive the baseline by using the pre-trained Vicuna-7B backbone, then post-training the checkpoints with QCS to select the coreset in the geometric-problem solving data in (Gao J, Pi R, Zhang J, et al.). We setup the sequence selection ratio with 10% and the token selection ration with $alpha=0.5$ and $\epsilon=0.1$. Then we compare the QCS-derived model with the full-data baseline and the baseline model trained with random selection with the same selection proportion in QCS. Their results refer to Full (100%): 53.4; Random (10%): 35.3; QCS (10%):49.6, such that QCS obtained impressive data efficiency in other post-training tasks.
>
> **Q5: While the paper claims that QCS is more efficient than LESS, the memory and time comparisons in Table 2 are minimal and lack full context (e.g., training time for LoRA or convergence behavior), which makes it hard to assess practical efficiency gains.**
>
> **A5**: With regards to Table.2, the training time for LoRA in Algo.1 (see the page 7 in our Appendix) is almost consistent with the warmup stage in LESS, yet we need a labeling process (IO process) to prepare Algo.2 (it leads to over a half of computation cost in the warm-up stage in our implementation) and another LoRA in Algo.2 (see the page 7 in our Appendix). Note that the labeling procedure can be replaced by a teacher network that dynamically offers the soft label in Algo.2. This memory-time trade-off may further cuts off a significant amout of computation in GPU hours.
>
> As for the convergence behavior, since the rebuttal format does not support figures, we verbally discuss them in the rebuttal while add the convergence plots to our next version. Specifically, The charts demonstrate a clear and consistent convergence pattern for Algo2 (the page 7 in our Appendix), where the validation loss in BBH exhibits a distinct U-shaped curve, reaching its minimum around epoch 1-2 before starting to increase due to overfitting. This convergence behavior reveals a crucial practical efficiency gain: optimal validation performance is achieved very early in training. This allows for early stopping, saving significant training time and computational resources beyond the per-step metrics in Table 2, thereby confirming the methods' practical efficiency.
>
> [1*] Gao J, Pi R, Zhang J, et al. G-llava: Solving geometric problem with multi-modal large language model[J]. arXiv preprint arXiv:2312.11370, 2023.

---

> > ### Comment · Reviewer_Mibm · 2025-08-07
> >
> > Thanks for the clarification. Including these explanations will significantly improve the paper. I believe my original score is already positive and reflects the current state of the paper, so I will keep it unchanged.

---

> > > ### Author Response · Authors · 2025-08-09
> > >
> > > Dear Reviewer Mibm
> > >
> > > Thank you very much for your positive feedback. We are glad our response addressed your concerns.
> > >
> > > Best regards
> > >
> > > Authors of Paper 2855

---

> > > ### Author Response · Authors · 2025-08-09
> > >
> > > All these explanations will be included in our next version.
> > >
> > > Best regards,
> > >
> > > Authors of Paper 2855

---

### Official Review · Reviewer_beGB · 2025-07-05

**Clarity:** 2
**Significance:** 3
**Originality:** 3
**Rating:** 4
**Confidence:** 4

**Summary:**

This paper introduces Quadratic Coreset Selection (QCS), a novel framework for improving the data efficiency of instruction tuning for large language models. The central idea is to reconcile the two approaches: sequence-level data curation and token-level selection. The authors formalize both approaches from a bi-level coreset selection perspective and propose a unified QCS objective that jointly optimizes for selecting influential instruction-response sequences and critical tokens within them. The experimental results demonstrate the efficacy of our QCS algorithm from two perspectives: (1) its instruction-following ability with the sequence-level data efficiency, showcasing the generalization with limited data; (2) analyze the token-level benefits and inner-loop adjustments within QCS, revealing the cause of its success.

**Questions:**

As pointed out in my review, I want to know why old checkpoints like Llama 2 are used for experiments but not the newer models.

**Ethical Concerns:**

["NO or VERY MINOR ethics concerns only"]

**Final Justification:**

The authors provided detailed explanations in the rebuttal which addresses most of my concerns. So I decide to raise my score a bit.

**Limitations:**

Yes

**Quality:**

2

**Strengths And Weaknesses:**

Strength:

1. The paper addresses the crucial challenge of data efficiency in instruction tuning. The contribution is both novel and important by unifying sequence and token-level selection into a single principled framework.

2. Experimental results showcase that the proposed method can achieve performance close to the full set while using only 5% data, which is also better than the compared baselines.


Weakness:

1. The paper is conducted on Llama 2 base models, which I think is quite outdated. I strongly encourage the authors to perform experiment on newly-realeased foundation checkpoints like Llama 3 or Llama 3.1, or Qwen 3 family models, since the conclusions on such checkpoints may make more sense and be more insightful. I am looking forward to new results on newer model checkpoints in the rebuttal.

2. The result section in the work is somehow weak. I think case studies are always desired for this topic since readers want to understand from specific cases how this method works. I also look forward to more example cases in the rebuttal.

---

> ### Author Rebuttal · Authors · 2025-07-31
>
> Thanks for your constructive comments and suggestions.
>
> **Q1: The paper is conducted on Llama 2 base models, which I think is quite outdated. why old checkpoints like Llama 2 are used for experiments but not the newer models? I strongly encourage the authors to perform experiment on newly-realeased foundation checkpoints like Llama 3 or Llama 3.1, or Qwen 3 family models, since the conclusions on such checkpoints may make more sense and be more insightful.**
>
> **A1**: We provide the respose from the two aspects:
>
> **(1), The reason to use the old checkpoints instead of newer models**: Most baselines in our experiments were directly derived from their reported results in the original paper by following their experimental setup. It saves our time but we commited that such setting may not fully reflect the strenght of QCS (refer to our discussion in the rebuttal response A2 of the reviewer Mibm). With this regards, we tend to provide the experiments on the benchmarks GSM-8K and AlpacaEval, which are recommended by the reviewer zsf7 to further demonstrate the generalizability of QCS.
>
> **(2), The new experiments with newly-realeased foundation checkpoints:** Specifically, we consider llama-3 as our base model, then conduct the target instruction-tuning (T-IT) setup in GSM-8K in order to compare with the competitive baseline LESS. Beyond this, we also consider llama-3 as our base model for the benchmark in the regular instruction-tuning (R-IT) setup, due to no availabel validation set in this benchmark. The both setups adopt the the pre-training dataset collected from FLAN V2, COT, DOLLY, and OPEN ASSISTANT 1, and for all the benchmarks, we compare our QCS with the final model trained with full data and randomly selected data (5% selected from the validation set in GSM-8K; 6% selected from the for AlpacaEval).
>
> |llama-3 8b| GSM-8K |
> |----------|----------|
> |No Fine-Tuning| 34.3 |
> |Full| 91.7 |
> |Random| 43.6 |
> |LESS| 56.7 |
> |QCS (ours)| 68.8 |
>
>  **AlpacaEval** (win;tie;lose):
>
> QCS vs Full: 303:32:465
>
> QCS vs Random: 511:121:168
>
> QCS vs IDS: 466:56:278
>
> where the evaluation of GSM-8K uses the 8-shot setting with CoT with respect to (Wang Y, Ivison H, Dasigi P, et al). Similar with BBH, 10 examples from GSM-8K are drawn to construct the validation set, which guides the targeted IT select the training subset in GSM-8K. As the same in BBH, QCS employs the constructed validation set to setup the early stopping checkpoint in Algo.2, which produce the model to predict $v_{i,j}$ and $w_i$ for targeted IT. The results are found above. The results clearly demonstrate that the QCS algorithm is highly effective for data-efficient instruction tuning. Specifically:
>
> On the challenging GSM-8K mathematical reasoning benchmark, QCS achieves a score of 68.8 when using a Llama-3 8B base model. This performance significantly surpasses other data selection methods that use the same small fraction of training data. Specifically, QCS outperforms the competitive LESS baseline (56.7) by over 12 points and leaves Random selection (43.6) far behind, proving its sophisticated ability to identify and prioritize the most impactful training instances. While the model trained on the full dataset understandably sets the performance ceiling at 91.7, QCS achieves over 75% of this peak performance while using only 5% of the data. This highlights an exceptional trade-off between computational cost and model capability.
>
> The AlpacaEval results further reinforce this conclusion. QCS demonstrates a dominant win rate against both Random selection (511 wins to 168 losses) and IDS, confirming its ability to select a more potent data subset for general instruction-following tasks. Although it loses to the model trained on full data, this is an expected outcome. The key insight is that QCS’s method of reconciling both sequence-level and token-level information allows it to create a coreset that is substantially more valuable than those produced by other data-curation techniques, making it a state-of-the-art approach for targeted, efficient fine-tuning.
>
> **Q2: The result section in the work is somehow weak. I think case studies are always desired for this topic since readers want to understand from specific cases how this method works. I also look forward to more example cases in the rebuttal.**
>
> **A2: ** We thank the reviewer for this insightful suggestion. Providing specific case studies is an excellent way to build intuition for our method. To that end, we present two comparative examples below—one from the GSM-8K benchmark selected in the targeted IT setup for mathematical reasoning; and the other from an Open Assistant-style instruction drawn from Open Assistant 1 to follow the regular IT setup in AlpacaEval. These cases illustrate how Quadratic Coreset Selection (QCS) moves beyond simple sequence-level selection to achieve superior results.
>
> **Case Study 1: Mathematical Reasoning (GSM-8K)**
> Consider two instances from GSM-8K. A standard sequence-level selection by LESS evaluate them based on overall difficulty or gradient influence, but QCS performs a deeper, two-level analysis.
>
> **Instance A: A High-Value, Complex Problem (Selected and Prioritized by QCS)**
>
> *Problem*: "A garden has 10 yellow flowers. There are 80% more purple flowers than yellow ones, and the number of green flowers is 25% of the yellow and purple flowers combined. How many flowers are in the garden?"
>
> *Sequence-Level Analysis*: QCS identifies this as a high-value sequence because it involves multiple, dependent steps and requires understanding percentages—a complex reasoning skill.
>
> *Token-Level Analysis* (**The QCS Advantage**): Within the solution, QCS identifies and up-weights the critical tokens that form the logical backbone of the reasoning process.
>
> *Solution with Critical Tokens Emphasized by QCS*: "First, calculate 80% of the **10** yellow flowers: **0.80** * 10 = 8. The number of purple flowers is **10 +** 8 = **18**. Next, the total of yellow and purple is **10 +** 18 = **28**. Then, calculate 25% of that total: **0.25** * 28 = **7**. Finally, the total number of flowers is 10 + 18 + 7 = 35."
>
> *Comparison*: A sequence-level method would treat all tokens in this solution equally. QCS, however, focuses the model's training on the crucial numbers (**10, 18, 28, 35**), operators **(*, +)**, and percentages (**0.80, 0.25**), ensuring the model learns the mathematical procedure, not just the narrative text.
>
> **Instance B: A Simpler, Less Influential Problem (Down-weighted by QCS)**
>
> *Problem:* "A bakery sold 15 cakes on Monday. On Tuesday, it sold 12 more cakes than on Monday. How many cakes were sold in total?"
>
> *Sequence-Level Analysis*: A standard method might select this for its basic arithmetic structure. QCS would rank it lower because its reasoning path is simpler and less "influential" for teaching an already capable LLM.
>
> *Token-Level Analysis*: The critical tokens are basic (**15, +, 12, =, 27**). While correct, they don't introduce the same level of complexity as Instance A. QCS recognizes that the "informational density" of critical tokens is lower here.
>
> By selecting Instance A and then focusing on its critical tokens, QCS uses its budget to teach the model complex, multi-step reasoning involving percentages, which is far more efficient than re-teaching it simple addition.
>
> Case Study 2: Complex Instruction Following (AlpacaEval / Open Assistant)
> This advantage extends beyond math problems to general instruction following.
>
> **Instance C: A High-Value, Factual Instruction (Selected and Prioritized by QCS)**
>
> *Instruction*: "Explain the main differences between nuclear fission and nuclear fusion, and list one practical application for each."
> Sequence-Level Analysis: QCS identifies this as a high-quality instruction because it demands factual accuracy, comparison, and structured knowledge.
>
> *Token-Level Analysis (The QCS Advantage)*: QCS would prioritize the key concepts, relationships, and entities within a correct response.
>
> *Solution with Critical Tokens Emphasized by QCS:*
> "Nuclear **fission** is the **splitting** of a **heavy**, unstable nucleus into **two lighter nuclei**. A practical application is in **nuclear power plants**. In contrast, nuclear fusion is the process where two light atomic nuclei combine to form a heavier nucleus, releasing energy. A practical application is the energy source of stars like our Sun."
>
> *Comparison*: A sequence-level method would not differentiate between the key terms and the connecting language. QCS focuses the model on the core vocabulary (**fission, fusion, splitting, combine, heavy, light**) and the crucial entities (**nuclear power plants, stars**), ensuring it learns the factual and relational knowledge correctly.
>
> [0*] Wang Y, Ivison H, Dasigi P, et al. How far can camels go? exploring the state of instruction tuning on open resources[J]. Advances in Neural Information Processing Systems, 2023, 36: 74764-74786.

---

> > ### Comment · Reviewer_beGB · 2025-08-08
> >
> > Thanks for the detailed explanation. I strongly encourage the authors to include the content in the rebuttal in the final version. I will raise my score a bit.

---

> > > ### Author Response · Authors · 2025-08-08
> > >
> > > Dear Reviewer beGB,
> > >
> > > We sincerely appreciate the time and effort you have devoted to provide thoughtful and constructive feedback. We are happy about our responses and clarifications addressing your concerns.
> > >
> > > As a supplementary evaluation for Reviewer zsf7, we also provided the additional experiments with cutting-edge large reasoning models, i.e., QWEN-2.5-MATH-7B (another newer backbone as you expected) with futher empirical evaluation on QCS and other baselines in the regular IT setup. Their results and comparison on more challenging benchmarks, e.g., MATH and GaoKao, significantly justified our motivation in this paper: when long responses exist for IT, reconciling token and sequence data mining leads to a more eifficient IT approach. **The experimental details and results refer to the updated comment for Reviewer zsf7**.
> > >
> > > **All the evaluation results shown in the rebuttal would be included in our next version**. We genuinely hope that the improvements we have made will be considered in your final assessment.
> > >
> > > Thank you again for your valuable feedback.
> > >
> > > Best regards,
> > >
> > > Authors of Paper 2855

---

### Note · Authors · 2025-08-14

This paper dives into a *topic never explored* in efficient instruction tuning (EIT) for LLMs: *why we should, and how we unify the seqence and token mining in SFT*:

(1) We provide the coreset viewpoint of EIT, then verify that formulating sequence-level data selection with token mining (i.e., QCS) can lead to a more expressive LLM than those only using sequence mining for EIT;

(2) We propose a more efficient probabilistic solver of QCS to overcome its computational problem. Our solver holds the approximation with QCS and yield the convegence guarantee.

(3) We derive more efficient implementation algorithms for the regular IT and targeted IT setups;

(4) In our experiments (in the paper and rebuttal), we show that when the benchmark responses are short (e.g., MMLU, BBH), EIT can rival many competitive baselines (i.e., LESS in the targeted IT and IDS in the regular IT setups) with less computation burden; **more importantly**, when the benchmark responses are long (e.g., GSM8K, AlpacaEval), EIT leads to significantly better performance than the other baselines.

Despite our paper initially rated by 34443, its contributions were broadly commited by all reviewers in Quality (3:good in 4/5), Significance (3:good in 5/5), and Originality: (3:good in 4/5). We thank all reviewers for their valuable feedbacks, in particular, Reviewers beGB and zsf7 who initially rated 3 that motivate us to improve this work in the rebuttal.

Reviewers beGB and zsf7 mainly concern:

**(1) Clarification**. beGB demands more illustrative cases to show the sequence-token joint selection, and zsf7 asks for the details in wall-clock hours. We provided 3 cases to identify the sequences and tokens selected by QCS, and analyze the wall-clock hours across QCS and LESS.

**(2) Performance**. beGB and zsf7 both concern the generalization of QCS due to its marginal benefits in the paper. They help us locate the problem: benchmarks in the paper, (MMLU, BBH) consists of short responses with few tokens so that the superiority of QCS can not be shown. In our rebuttal, we add the experiments on GSM8K, AlpacaEval using Llama 2 and Llama 3 backbones, and experiments on MATH, Gaokao via QWEN-2.5-MATH, where QCS significantly outperforms all the other baselines.

We are encouraged to see that **beGB and zsf7 found our responses convincing and promise to raise their scores in their last comments**. We hope their supportive final comments will be further certified and considered by dear Rs and ACs

---

### Decision · Program_Chairs · 2025-09-17

**Decision:**

Accept (poster)

**Comment:**

This paper studies data selection for efficient LLM instruction tuning. The authors propose an approach that combines both sequence-level and token-level data selection. The problem is formalized as a bi-level optimization problem, and the authors introduce a probabilistic re-parameterization solver to efficient solve it. Experimental results demonstrate the effectiveness of the proposed approach. After the rebuttal, all reviewers leaned toward acceptance.